# Inter-individual and inter-site neural code conversion without shared stimuli

Haibao Wang [1,2,3] ✉, Jun Kai Ho [1,2], Fan L. Cheng [1,2], Shuntaro C. Aoki [1,2], Yusuke Muraki[1], Misato Tanaka [1,2], Jong-Yun Park [2] & Yukiyasu Kamitani [1,2] ✉

Inter-individual variability in fine-grained functional topographies poses challenges for scalable data analysis and modeling. Functional alignment techniques can help mitigate these individual differences but they typically require paired brain data with the same stimuli between individuals, which are often unavailable. Here we present a neural code conversion method that overcomes this constraint by optimizing conversion parameters based on the discrepancy between the stimulus contents represented by original and converted brain activity patterns. This approach, combined with hierarchical features of deep neural networks as latent content representations, achieves conversion accuracies that are comparable with methods using shared stimuli. The converted brain activity from a source subject can be accurately decoded using the target's pre-trained decoders, producing high-quality visual image reconstructions that rival within-individual decoding, even with data across different sites and limited training samples. Our approach offers a promising framework for scalable neural data analysis and modeling and a foundation for brain-to-brain communication.

Individual differences in brain organization are observed at different scales, from macroscopic anatomy to fine-grained functional topography[1–6]. These individual differences may complicate and confound group-level analyses and make it challenging to generalize individual-specific models to other individuals. Although anatomical alignment can help mitigate anatomical differences[1–3], it does not fully account for the variability in functional topography across individuals, probably due to idiosyncratic neural representations at finer-grained scales[4–6].

Functional alignment has been important in functional magnetic resonance imaging (fMRI) research, addressing individual differences in functional topography. This approach generally involves presenting the same stimuli to different subjects and aligning brain activity patterns to make them similar or shared across individuals[5–11]. Functional alignment encompasses two primary strategies: pairwise and template-based alignments. The former, which include neural code converters[8,11], convert the brain activity pattern of one subject into that of another, representing the same content. The latter, which include

hyperalignment[5–7,9], create a common brain activity space for all subjects. However, functional alignment typically requires identical stimuli (shared stimuli) to be presented to different individuals to obtain paired brain data for training; such stimuli are often unavailable and limit its broader application.

Although shared stimuli in functional alignment ensure that brain activity patterns across individuals reflect the same stimulus content, the content can be more flexibly represented by a combination of elemental, latent features, such as image bases or deep neural network (DNN) features[12–14]. By decomposing a single stimulus into these elemental features, functional alignment may encompass a broader range of stimuli beyond those that are explicitly shared. Recent advancements in brain decoding research have revealed that the contrasts of local image bases and the latent features from DNNs can be decoded from the brain activity patterns measured by fMRI[15,16]. Furthermore, these decoded features can be transformed into images by linear or deep generative networks[15,17–19], reconstructing perceptual experiences. These results suggest the potential for using latent

[1]Graduate School of Informatics, Kyoto University, Kyoto, Japan. [2]Department of Neuroinformatics, ATR Computational Neuroscience Laboratories, Kyoto, Japan. [3]Guardian Robot Project, RIKEN, Kyoto, Japan. ✉e-mail: haibaowa@gmail.com; kamitani@i.kyoto-u.ac.jp

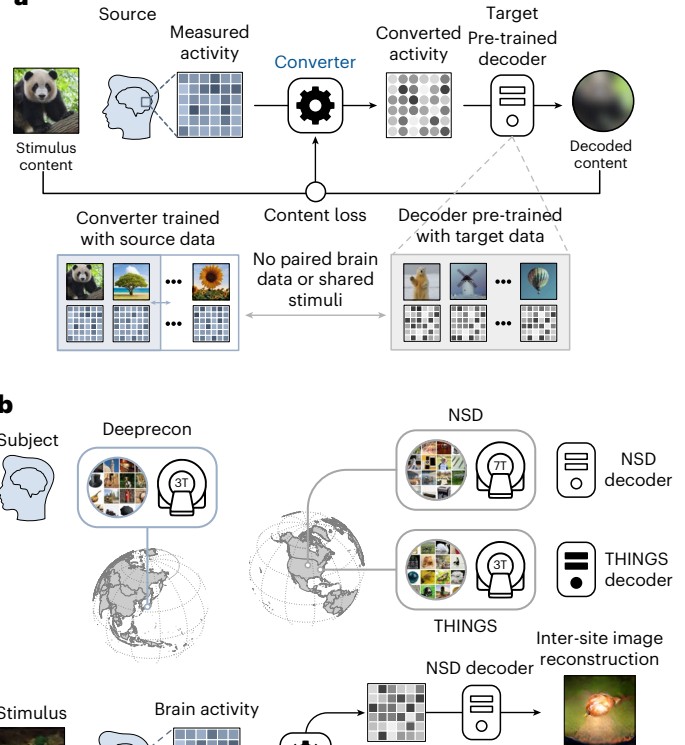

**a**

**b**

**Fig. 1 | Content-loss-based neural code conversion and inter-individual (-site) image reconstruction. a**, Training the neural code converter with content-loss-based optimization. The target subject's training data are used to pre-train the target decoder, whereas the source subject's training data are used for converter training. The converter is optimized so that the converted brain activity is decoded into content representations that closely resemble those of the stimulus given to the source subject. During the training stage, there was no need for paired brain data or shared stimuli between the source and target. **b**, The example of inter-individual (-site) visual image reconstruction. The trained converter functions across different datasets, which may have different stimuli and scanners with different resolutions. When presented with a new stimulus, the subject's brain activity pattern from one dataset is converted into the space of another dataset. The converted pattern is then decoded into image feature representations by a feature decoder within the target dataset, enabling inter-individual (-site) image reconstruction as perceived by the source subject. Some original stimulus images have been replaced for copyright reasons.

features instead of shared stimuli to ensure content similarity in functional alignment.

In this study we introduce a content-loss-based functional alignment method that converts brain activity patterns between individuals without requiring shared stimuli. For each subject pair, we designate one as the target and the other as the source. First, we pre-train decoders to predict the latent features of stimuli perceived by the target subject on the basis of their measured brain activity[16]. A neural code converter model is then optimized to minimize the content loss between the latent features of the stimulus and those decoded from the converted brain activity (Fig. 1a). This training approach does not rely on paired brain activity (shared stimuli), allowing the source and target subjects to originate from different datasets or without any shared stimuli. In the test stage, the converter transforms the source subject's brain responses to new stimuli into the target's brain space. These converted patterns can then be decoded and used to reconstruct images, reflecting the extent to which visual information is preserved

across individuals. This approach works not only between individuals but also across different experimental sites (Fig. 1b), allowing brain data from one site to be converted and decoded using models from another site.

We evaluate the neural code conversion approach through several key analyses. First, we assess the method's ability to convert brain activity patterns across individuals and capture fine-grained visual features for inter-individual image reconstruction, by comparing converters optimized with visual content versus paired brain activity. We then examine whether the method remains effective without shared stimuli, by evaluating conversion performance with both overlapping and non-overlapping stimuli between converter and decoder trainings. To demonstrate broader applicability, we examine inter-site conversion and reconstruction across different fMRI datasets collected under varying experimental conditions. We also assess the generalizability of converted brain activity using an alternative decoding scheme and evaluate the method's performance with limited training data. Although the method is primarily validated in the visual domain, we conduct a preliminary analysis to explore its potential extension to other cognitive domains. Through these analyses, we aim to establish a flexible framework for functional alignment that eliminates the need for shared stimuli while preserving fine-grained neural representations.

## Results

### fMRI data and content representations

Our analyses used three fMRI datasets for natural images, involving 114 subject pairs total (refer to the 'Datasets' section in the Methods). The primary analysis used the Deeprecon dataset[11,18,20] with five subjects, each having 6,000 training and 2,000 test samples, yielding 20 subject pairs. For inter-site analysis, we incorporated two additional datasets: the THINGS dataset[21] with three subjects (8,640 training and 1,200 test samples each) and the Natural Scene Dataset[22] (NSD) with four subjects (around 25,000 training and 300 test samples each), creating 94 inter-site pairs. For the training of neural code converters, we used the hierarchical features from the VGG19 model[13] as content representations (Fig. 2a,b; refer also to the 'DNN models' section in the Methods), analyzing fMRI responses from the whole visual cortex (VC). To explore domains beyond vision, we also analyzed fMRI responses to natural sounds in the auditory cortex using the DeepSoundRecon dataset[23] and the sound features extracted from the VGGish-ish DNN model[24] as content representations. Unless otherwise noted, all decoding and reconstruction analyses used test samples averaged across repetitions for each stimulus.

### Neural code conversion

We first investigated whether brain activity patterns could be reliably aligned across individuals using a content-loss-based converter within the Deeprecon dataset. The conversion was performed across the entire visual cortex, and its accuracy was evaluated by two metrics: (1) pattern correlation, which is the Pearson correlation coefficient between the converted and measured voxel patterns for a test stimulus (Fig. 2c), and (2) profile correlation, which is the Pearson correlation coefficient between the sequences of converted and measured responses of a single voxel to the 50 natural test stimuli (Fig. 2d), both normalized against noise ceilings (refer to the 'Evaluation of conversion accuracy' section in the Methods). To summarize the results, we averaged the pattern correlation coefficients across all stimuli and profile correlation coefficients across all voxels within each individual pair and each region of interest (ROI). For comparison, we also evaluated the brain-loss-based neural code converter[8,11] (Fig. 2b; refer to the 'Methods for functional alignment' section in the Methods).

Figure 2e,f presents the distributions of pattern correlation coefficients and profile correlation coefficients across all conversion pairs for different ROIs in the target brain space (see Supplementary Fig. 1 for results of individual pairs). Although the content-loss-based converter

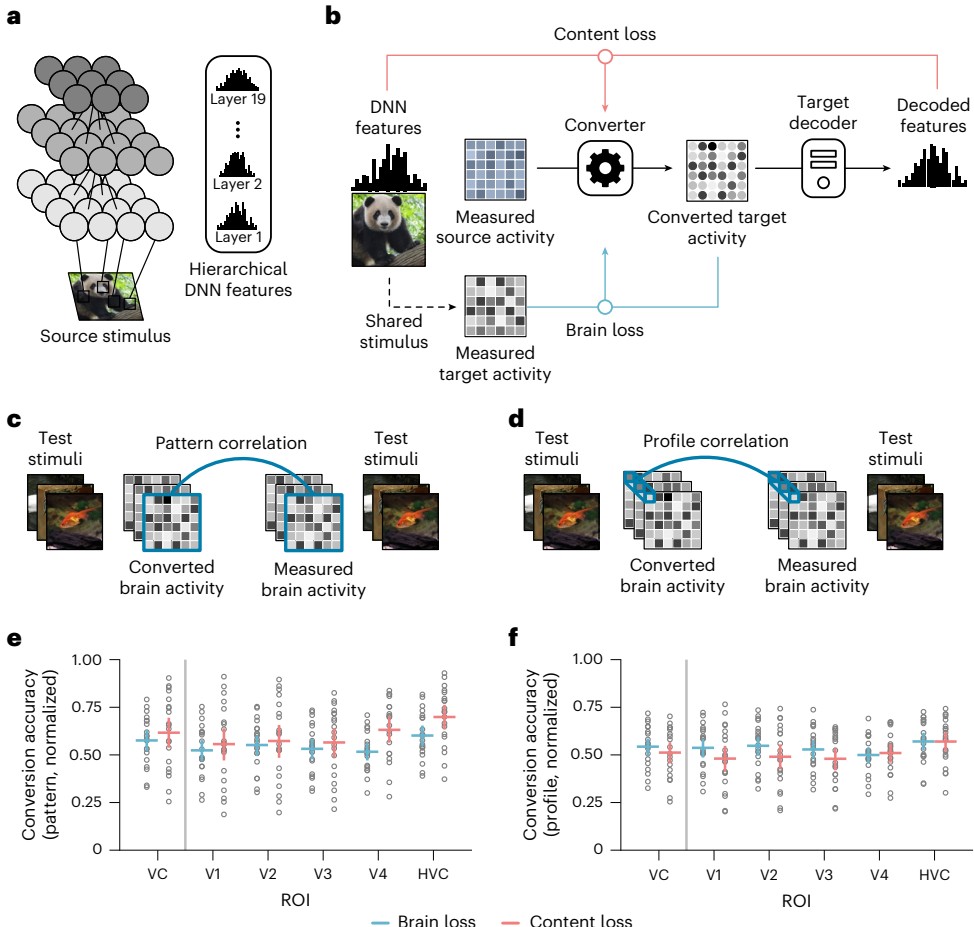

**Fig. 2 | Neural code conversion using content and brain losses. a**, Hierarchical DNN features as visual contents. **b**, The overview of converter training using content- and brain-loss-based optimization. The former minimizes the loss between the DNN features decoded from the converted target activity and those extracted from the corresponding stimulus; the latter minimizes the loss between the converted target activity and the measured target activity. **c**, Evaluation of neural code conversion (pattern). The Pearson correlation coefficient between the converted and measured voxel patterns for a test stimulus was calculated for pattern correlation. **d**, Evaluation of neural code conversion (profile). The Pearson correlation coefficient between the sequences of converted and measured responses of a single voxel to the 50 natural test stimuli was calculated for profile correlation. **e**, Conversion accuracies measured by pattern correlation. Distributions of the pattern correlation coefficients of 20 pairs are shown for the whole visual cortex and its subareas, including V1, V2, V3, V4 and the higher visual cortex (HVC). The horizontal bars represent the mean accuracies across the 20 pairs, with vertical bars representing the 95% CI, and each dot representing an individual pair's mean correlation coefficient over stimuli. **f**, Conversion accuracies measured by profile correlation. The two converters are compared as in **e** but with profile correlation coefficients.

was not explicitly trained to align brain activity patterns, its performance measured by correlations between converted and measured brain activity patterns was comparable with—or only slightly lower than—the brain-loss-based method across all examined visual subareas. Although some differences in variance are observed between the two methods, they were not statistically significant (post-hoc Levene's test, pattern correlation, $P > 0.05$ for all ROIs except V2, where $P = 0.039$; profile correlation, $P > 0.05$ for all ROIs). Both methods demonstrated relatively consistent alignment, with individual pairs showing correlated performance between the two methods (Supplementary Fig. 2). We also tested other standard methods for functional alignment (hyperalignment[6]), showing similar conversion accuracies (see Supplementary Fig. 3 for further details on functional alignment methods). The analysis of converters trained using the content loss from different DNN layers revealed that the use of multiple DNN layers contributes to accurate neural code conversion (Supplementary Fig. 4).

We also extended our analysis to different selections of visual ROIs. We performed subarea-wise conversion analyses by training separate converters for paired ROIs between source and target subjects[11]. When using corresponding ROIs, the subarea-wise approach achieved accuracies that were comparable with the whole VC conversion approach (see Supplementary Fig. 5 for further details). When predicting target ROI activity from different source ROIs, the highest conversion accuracies were observed between subareas at similar hierarchical levels (Supplementary Fig. 6). These results indicate that the content-loss-based approach can effectively convert brain activity patterns across individuals, preserving hierarchical neural representations.

To examine whether the content loss approach remains effective without shared stimuli, we performed neural code conversion with no stimulus overlap between the converter and decoder trainings. Specifically, we randomly split the training samples of the Deeprecon dataset into two halves on the basis of stimulus categories, assigning one to the source and the other to the target to ensure no overlap in their training stimuli and categories (referred to as the non-overlapping condition; see Supplementary Fig. 7a for details). The conversion accuracy in the non-overlapping condition was comparable with that in the overlapping condition, in which the source and target share the same training stimuli (Supplementary Fig. 7), demonstrating the robustness of the content loss approach.

To further evaluate the potential of our approach, we conducted preliminary analyses examining neural responses in the auditory cortex during sound stimulus presentation[23]. Using the same

content-loss-based framework, we observed robust inter-individual conversion of auditory neural representations (Supplementary Fig. 8a–e). The performance trends were similar to those observed in the visual domain (Fig. 2e,f); however, profile correlations showed a greater decrease in the content-loss-based method. The successful conversion of both visual and auditory cortical responses suggests the potential versatility of our approach for studying neural representations across different sensory and cognitive domains.

### Inter-individual DNN feature decoding

We next investigated whether fine-grained feature representations of stimuli were preserved in the converted brain activity by a DNN feature decoding analysis[16] (Fig. 3a; refer to the 'DNN feature decoding analysis' section in the Methods). We evaluated the decoding accuracy using two metrics: (1) pattern correlation, which is the Pearson correlation coefficient between the pattern of decoded features and the true feature patterns for a test stimulus, and (2) profile correlation, which is the Pearson correlation coefficient between the sequences of the decoded and true feature values across all test stimuli for a single DNN unit. We further averaged the pattern correlations across all test stimuli and the profile correlations across all DNN units within each layer. For comparison, we also calculated the decoding accuracy for the standard within-individual decoding ('Within' condition), which predicts DNN features using decoders trained exclusively on data from the same subject.

The content-loss-based converters exhibited comparable decoding accuracies with those obtained through the within-individual decoding, with consistently similar trends across various DNN layers (Fig. 3b,c). The content-loss-based converters showed higher accuracy than those using brain loss regarding both pattern and profile correlations. Other standard functional alignment methods exhibited lower accuracies (Supplementary Fig. 9a,b). The decoding analysis of converters trained using the content loss from different DNN layers showed that converters trained with all DNN layers achieved better decoding accuracy (Supplementary Figs. 10 and 11). The content-loss approach yielded comparable decoding accuracy, whether using overlapping or non-overlapping stimuli between converter and decoder trainings (Supplementary Fig. 12a,b). Preliminary auditory decoding analyses within the whole auditory cortex revealed consistent performance for content-loss-based converters (Supplementary Fig. 8f,g). These results suggest that content-loss-based converters effectively preserve the fine-grained feature representations of stimuli.

### Inter-individual visual image reconstruction

We further reconstructed images using fine-grained DNN feature representations decoded from converted brain activity (Fig. 3a; refer to the 'Visual image reconstruction' section in the Methods) and showed examples of the reconstructions from the whole visual cortex (Fig. 3d,e; see Supplementary Figs. 13 and 14 for the reconstructions of all pairs; see Supplementary Figs. 15 and 16 for the reconstructions from each ROI). The reconstructed images obtained from converters using brain and content losses captured the essential characteristics and details of the presented images, with visual objects being similarly recognizable to those in the within-individual reconstructions. The other functional alignment methods showed similar reconstructions (Supplementary Fig. 9c).

We performed a pairwise identification analysis for a quantitative evaluation of reconstruction performance. This analysis used the pixel and DNN feature patterns of the reconstructed image to identify the true stimulus between two alternatives (refer to the 'Identification analysis' section in the Methods). We used the AlexNet model[12] to extract DNN feature patterns. The identification process was repeated with multiple false alternatives to obtain the accuracy for each reconstructed image. We then calculated the mean identification accuracy across all reconstructions for each individual pair, presenting these

accuracies at the group level in Fig. 3f,g. Features extracted from reconstructions achieved higher identification accuracy at the mid-level layers, whereas correlation-based evaluation of decoded features showed higher performance at the high-level layers (Fig. 3b,c). The correlation analysis of extracted features from reconstructions also showed higher accuracies at the high-level layers (Supplementary Fig. 17). This discrepancy between identification and correlation metrics can arise because successful identification requires features that distinctly differentiate between stimuli, whereas high correlations can occur even when predictions capture only common patterns across stimuli[16].

We observed that both types of inter-individual converters achieved comparable but slightly lower identification accuracies than within-individual reconstruction (Fig. 3f,g). In comparing the two inter-individual approaches, the content-loss-based converter demonstrated better performance than the brain-loss-based converter. The other standard functional alignment methods achieved slightly lower accuracies (Supplementary Fig. 9d). Different regression model architectures used for content-loss-based converters demonstrated comparable performances (Supplementary Fig. 18). The use of multiple hierarchical DNN layers in the content-loss-based converter shows reliable inter-individual reconstruction performance (Supplementary Fig. 19). The converter trained with non-overlapping stimuli achieves similar accuracy (Supplementary Fig. 12c,d). These results demonstrate that content-loss-based converters can convert brain activity patterns across individuals while preserving fine-grained neural representations, even without requiring overlapping (shared) stimuli.

### Inter-site neural code conversion and image reconstruction

We extended our analysis to investigate the feasibility of inter-site neural code conversion, where converters were trained between source and target subjects from distinct sites. For this analysis, we used the Deeprecon dataset and two additional datasets: the NSD and the THINGS dataset. The source subject was selected from one of these three datasets, whereas the target subject was selected from a different dataset, resulting in 94 individual pairs with no stimuli shared between any pairs. Due to the absence of ground truth (measured brain activity) for evaluating conversion accuracy in the brain space, we opted to evaluate the inter-site neural code conversion through the decoding accuracy and image reconstruction from the converted brain activity.

The inter-site decoding shows a similar decoding tendency and comparable accuracy across all DNN layers compared with the within-individual decoding for test samples from each dataset (Supplementary Fig. 20). Figure 4a–c presents examples of the inter-site image reconstruction (see Supplementary Figs. 21–23 for the reconstructions of 94 individual pairs). The inter-site reconstructions captured the core characteristics of the presented images, including their shape, color and textures, showing visual content similar to that of the within-individual reconstructions. A quantitative evaluation revealed that the identification accuracy of inter-site reconstructions was slightly lower but still comparable to within-individual reconstructions (Fig. 4d–f). These results further validated that our method enables inter-individual and inter-site neural code conversion and image reconstruction without shared stimuli.

### Generalizability of converted brain activity

To confirm whether the converted brain activity contains generalizable representations rather than those tailored for a specific readout (the VGG19 decoder), we examined the feasibility of decoding the converted brain activity using a different scheme. A different feature decoder was trained to predict CLIP-ViT features[14] (refer to the 'DNN models' section in the Methods) of the stimuli from the target subject's brain activity patterns. This decoder was then tested on the converted brain activity obtained from VGG19-based converters. We further reconstructed images from the decoded CLIP-ViT features using the same reconstruction method.

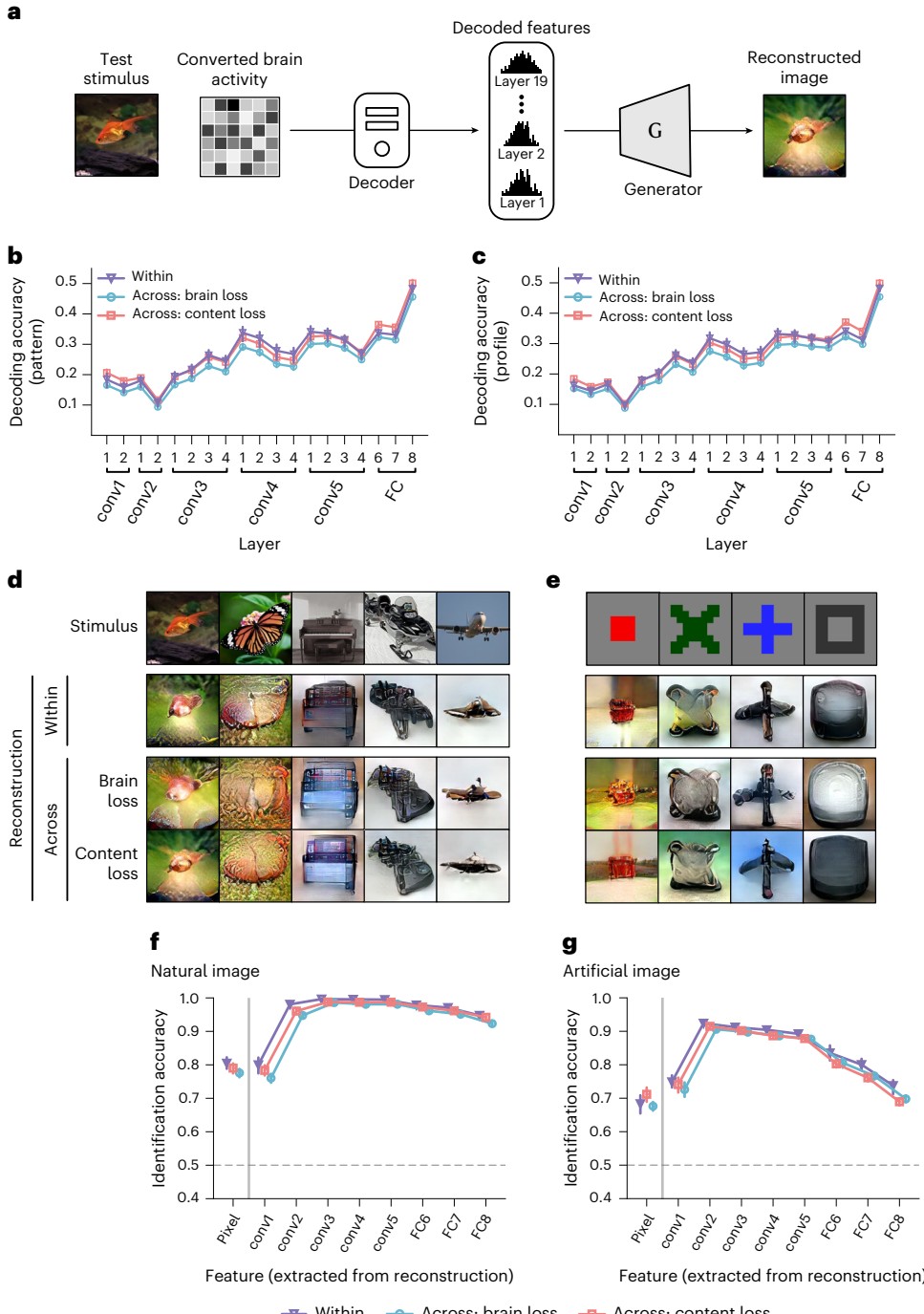

**Fig. 3 | Inter-individual decoding and image reconstruction. a**, DNN feature decoding and image reconstruction. The converted brain activity was decoded into DNN features using the decoder in the target space. The decoded features were fed into a reconstruction algorithm to reconstruct images. **b**, Feature decoding accuracy measured by pattern correlation. The DNN features are from the convolutional layers (conv) and fully connected layers (FC) of the VGG19 model. The mean pattern correlation for each layer is shown for the Within, brain-loss and content-loss conditions (VC; error bars, 95% CI from five subjects for the Within condition, and from 20 pairs for the brain-loss and content-loss conditions). **c**, Feature decoding accuracy measured by profile correlation. The three conditions are compared as in **b** but with profile correlation coefficients. **d**, Reconstructions of natural images. The reconstructions under the three analytical conditions for each stimulus image were all from the same source subject (VC; source, subject 1; target, subject 2; see Supplementary Fig. 13 for

the reconstructions across all individual pairs; see Supplementary Fig. 15 for the reconstructions from each ROI). Some stimulus images have been replaced with visually similar alternatives for copyright reasons. **e**, Reconstructions of artificial images. The reconstructions are shown as in **d** but with artificial images (see Supplementary Fig. 14 for the reconstructions across all individual pairs; see Supplementary Fig. 16 for the reconstructions from each ROI). **f**, Identification accuracy of natural images. Pairwise identification was performed using the pixel values and the extracted DNN feature values of the reconstructions. The DNN features were extracted from five convolutional (conv) layers and three FC layers of the AlexNet model. The mean identification accuracy was calculated over all reconstructed images for each subject or individual pair, and then averaged across all subjects or pairs to obtain the group-level mean (error bars, 95% CI from five subjects or 20 pairs; dashed lines, 50% chance level). **g**, Identification accuracy of artificial images.

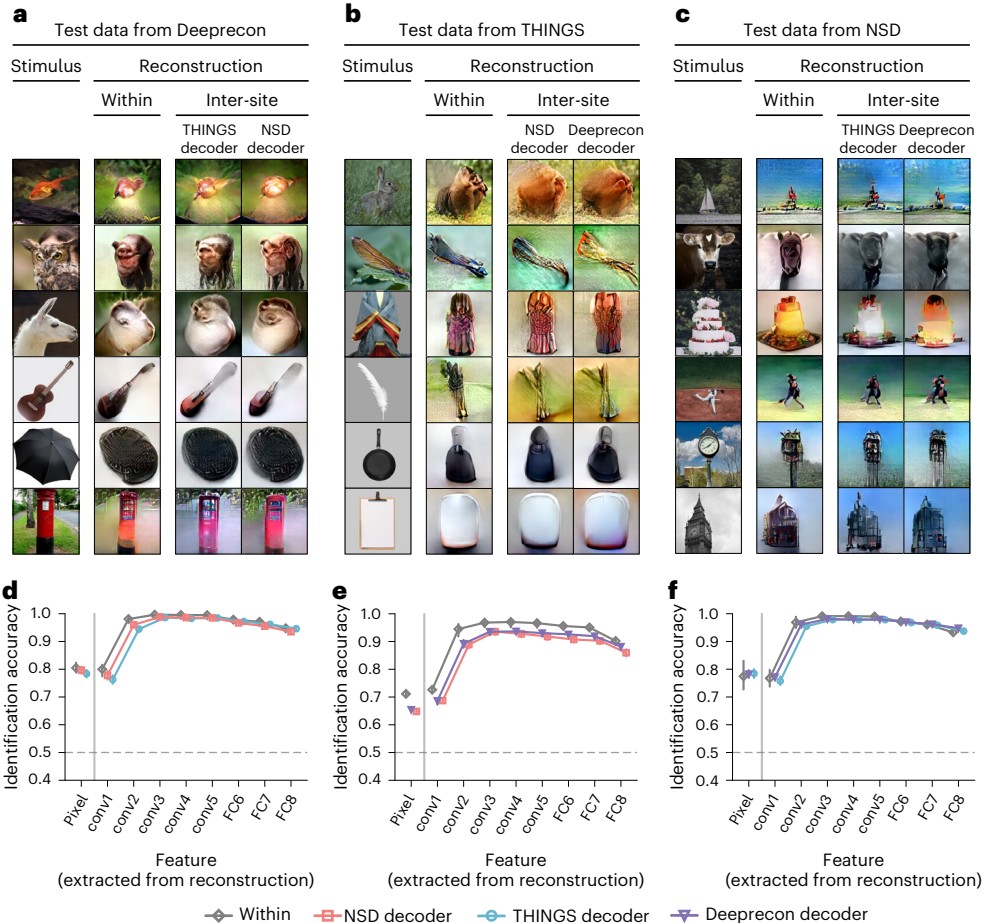

**Fig. 4 | Inter-site image reconstruction. a**, Reconstructions for Deeprecon test data. Within-individual reconstructions were obtained using decoders from the same subject, whereas inter-site reconstructions were obtained using the target's decoders from other datasets (VC; source, subject 1; target, subjects 6 and 9; see Supplementary Fig. 21 for reconstructions of all pairs). Some stimulus images have been replaced with visually similar alternatives for copyright reasons. A video illustrating more examples of the image optimization process is available at ref. 61. **b**, Reconstructions for THINGS test data. Examples for source subject 6 and target subjects 1 and 9 are shown (VC; see Supplementary Fig. 22 for reconstructions of all pairs; stimulus images were replaced for copyright reasons; refer to the video at ref. 62). **c**, Reconstructions for NSD test data. Examples for source subject 9 and target subjects 1 and 6 are shown (VC;

see Supplementary Fig. 23 for reconstructions of all pairs; stimulus images were replaced for copyright reasons; refer to the video at ref. 63). **d**, Identification accuracy for Deeprecon test data. Pairwise identification was performed using the pixel values and the extracted DNN feature values from the reconstructions. The DNN features were extracted from eight layers of the AlexNet model, including five convolutional layers and three FC layers. The mean identification accuracy was calculated across all subjects or individual pairs (error bars, 95% CI from five subjects or 20 pairs; dashed lines, 50% chance level). **e**, Identification accuracy for THINGS test data. The mean identification accuracy was calculated as in **d** but with test data from the THINGS. **f**, Identification accuracy for NSD test data. The mean identification accuracy was calculated as in **d** but with test data from the NSD.

Decoding CLIP-ViT features from converted brain activity demonstrates accuracies similar to those of the within-individual decoding across all evaluated DNN layers (Supplementary Fig. 24). Figure 5a shows examples of reconstructed images from these decoded CLIP-ViT features (see Supplementary Fig. 25 for the reconstructions of all pairs). The reconstructed contents resemble those obtained with the VGG19 decoder, although the details differ slightly. The quantitative evaluation showed that reconstructions with the CLIP-ViT decoder yielded identification accuracy approaching that of reconstructions with the VGG19 decoder across all evaluations (Fig. 5b). These results suggest that the converted brain activity is not tailored to the specific decoder and can be read out by different decoding schemes.

### Varying the number of training data for conversion
We investigated the effect of training sample size for converters on image reconstruction quality. We varied the number of training samples for converters (trained with VGG19 features) at various levels: 300, 600, 900, 1,200, 2,400, 3,600, 4,800 and 6,000 training samples, with data collection time ranging from 40 min to approximately 13 h. To

evaluate the decoding from converted brain activity, we consistently used 6,000 samples from the target subject to train the VGG19 and CLIP-ViT feature decoders.

Figure 6a shows the reconstructed images with the VGG19 decoder, using converters trained with varying sample sizes. Although the visual quality of the reconstructions diminished with smaller training sample sizes, converters trained with as few as 300 or 600 samples (around 1 h) still yielded perceptible images. The identification accuracy increased with the number of training samples, gradually approaching the accuracy observed in the within-individual reconstruction (Fig. 6b). Similar results were obtained for the reconstruction using the CLIP-ViT decoder (Supplementary Fig. 26). These results indicate the feasibility of inter-individual image reconstruction using converters trained on limited data, achieving modest performance while reducing reliance on extensive fMRI data.

### Discussion
We aimed to develop a functional alignment method that: (1) eliminates the need for shared stimuli across individuals during training; and (2)

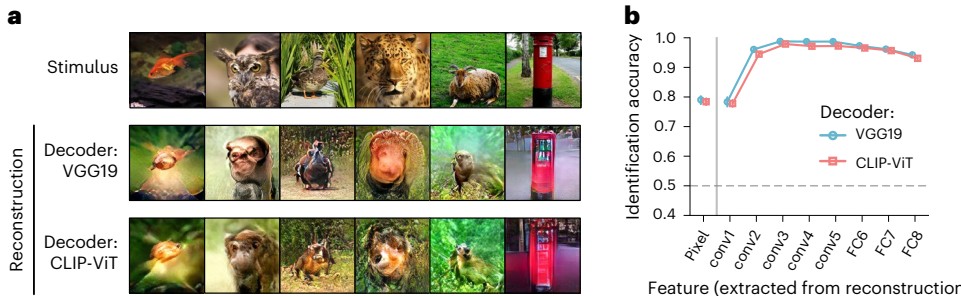

**Fig. 5 | Reconstruction by converter and decoder trained with different DNNs. a**, Reconstructed images using the VGG19 and CLIP-ViT decoders. The source brain activity was converted into the target's space using VGG19-based converters and then evaluated using the VGG19 and CLIP-ViT decoders of the target subject. The reconstructions were generated from the same pair (VC; source, subject 1; target, subject 2; see Supplementary Fig. 25 for reconstructions of all pairs). Some stimulus images have been replaced with visually similar alternatives for copyright reasons. **b**, Identification accuracy. Pairwise identification was performed using the pixel values and the extracted DNN feature values from the reconstructions obtained via the VGG19 or CLIP-ViT decoder. The DNN features were extracted from five convolutional (conv) layers and three FC layers of the AlexNet model. The mean identification accuracy was calculated across all individual pairs (error bars, 95% CI from 20 pairs; dashed lines, 50% chance level).

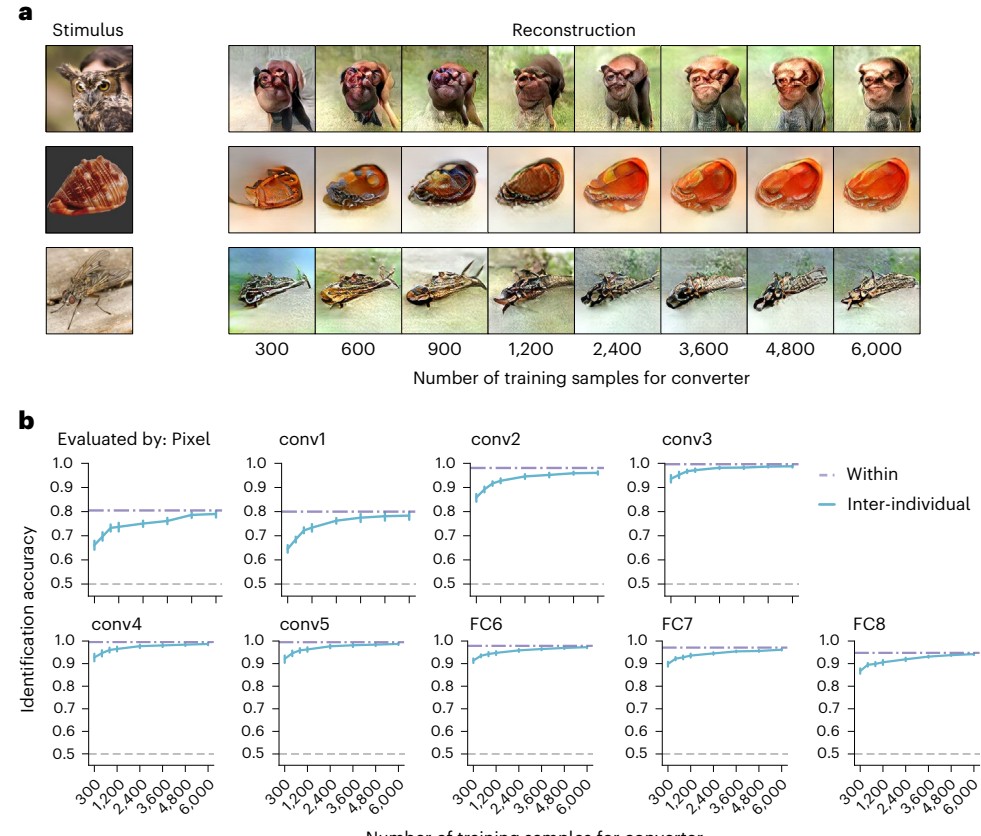

**Fig. 6 | The effect of the number of training data for conversion.**
**a**, Reconstructed images using the VGG19 decoder. Reconstructed images were generated with a varying number of training data from the same pair (VC; source: subject 1, target: subject 2; see Supplementary Fig. 26a for reconstructions using the CLIP-ViT decoder). Some stimulus images have been replaced with visually similar alternatives for copyright reasons. **b**, Identification accuracy. Pairwise identification was performed using the pixel values and the extracted DNN feature values from the reconstructions with varying quantities of training data. The DNN features were extracted from five convolutional (conv) layers and three FC layers of the AlexNet model. The mean identification accuracy was calculated across all individual pairs (error bars, 95% CI from 20 pairs; dashed lines, 50% chance level; see Supplementary Fig. 26b for identification accuracy using the CLIP-ViT decoder). The results are shown with those from the Within condition (6,000 training samples).

captures fine-grained feature representations, enabling downstream tasks such as decoding and image reconstruction. A major difference between our method and previous ones is the content-loss-based optimization, which minimizes the discrepancy between the stimulus contents and those decoded from converted brain activity patterns, rather than directly minimizing differences between paired brain activity patterns.

Although we used brain-loss-based approaches with shared stimuli, shared stimuli may not be strictly necessary in other contexts. For example, alignment can still be achieved without shared stimuli through methods such as connectivity hyperalignment[5,25], the Individualized Neural Tuning model[26], or predicted brain data[27]. Brain-loss-based approaches may also have limitations. Brain responses include not only consistent stimulus-evoked responses but also

idiosyncratic stimulus-evoked responses and noise components[28]. As a result, brain-loss-based methods may inadvertently incorporate these extraneous components, potentially reducing their effectiveness. Our content-loss-based approach may help mitigate this issue by optimizing stimulus representations.

The content-loss-based converter uses a training approach similar to feature decoders, but it specializes in learning the statistical relationship between the source and target brain spaces in several key aspects. First, it successfully converts brain activity patterns across individuals, achieving conversion accuracy similar to converters optimized by brain loss (Fig. 2). Second, the converted brain activity can be interpreted by another decoding scheme (Fig. 5), suggesting that the converter learns a more generalizable mapping between brain spaces. Finally, converter training requires fewer samples to achieve successful image reconstruction than feature decoder training (Fig. 6), suggesting a separate underlying mapping structure distinct from feature decoding.

Our primary goal was to convert brain activity across individuals' innate spaces without shared stimuli, enabling their use in downstream tasks such as decoding and image reconstruction. However, inter-individual decoding can still be achieved without explicit alignment. A multi-subject shared decoder can be trained with alignment layers optimized for specific tasks[29,30], implicitly creating a shared latent space for brain data. Similar approaches have been applied to decoding brain data from magnetoencephalography[31,32], electroencephalography[31] and electrophysiological neural recordings[33]. Although these studies have shown promising results for specific tasks, they do not aim to align brain data within the innate space, and the potential of converted brain data is not carefully examined. By contrast, our approach directly converts brain data into the target's innate space, enabling explicit evaluation of the conversion and seamless integration into existing analysis pipelines, as demonstrated by successful decoding and reconstructions (Figs. 2–6).

Although our approach demonstrates effectiveness, it is subject to dataset- and model-related limitations. The substantial overlap in semantic and visual content between training and test sets in the NSD dataset may increase the risk of spurious predictions[34] and introduce bias in inter-site reconstructions. Further, our model's reliance on VGG19 features as content representations may not fully capture the diverse aspects of visual experience, including motion perception, 3D perception and global perception.

The potential of our approach extends in several directions. First, determining the optimal features for visual content in neural code conversion requires further investigation. Future adaptations could use latent features derived from established models such as motion[35] and 3D perception[36] models to provide complementary insights. Furthermore, when combined with reconstruction models, pixel-level image representations can serve as visual content and directly define content loss. Second, although we demonstrate the effectiveness of our method in both visual and auditory tasks, its broader applicability to other tasks such as decision-making, language processing and reinforcement learning-based cognitive tasks, requires further investigation. Latent representations from these tasks, optimized using neural networks, may also serve as a basis for functional alignment. Third, our method is adaptable to brain data from various modalities. Although we primarily analyze brain activity patterns from fMRI signals, the content loss approach can be extended to electroencephalography and magnetoencephalography, enabling inter-modality decoding.

Our approach has practical implications in several aspects. First, our approach enables brain data conversion across datasets, expanding the reuse of publicly available data and supporting scalable analysis and modeling beyond institutional and geographical barriers. Second, our approach potentially reduces both the economic costs and time investments required for brain data collection by inter-individual decoding with fewer training samples (Fig. 6). Third, as indicated by the conversion between different visual areas (Supplementary Fig. 6), our approach could provide a data-driven framework for exploring functional homology between brain regions by comparing neural representations. It may help explore cross-species homologies by identifying regions that process information in similar ways, even when their anatomical locations or evolutionary origins differ. Finally, it may facilitate brain-to-brain communication through high-resolution stimulation technologies[37,38]. The converted brain activity preserving fine-grained information can be used to design high-resolution brain stimulation patterns and to induce mental content similar to that of the source subject. This enables the transmission of mental content between two individuals, providing a foundation for brain-to-brain communication.

## Methods
### Datasets
We reanalyzed four previously published and publicly available datasets[11,18,20–23]. In this study we used 16 subjects from these four datasets. Subjects 1–5 are sourced from the Deeprecon dataset[11,18,20], subjects 6–8 correspond to subjects 1–3 from the THINGS dataset[21], subjects 9–12 correspond to subjects 1, 2, 5 and 7 from the NSD dataset[22], and subjects 13–16 correspond to subjects 2–5 from the DeepSoundRecon dataset[23]. We used the preprocessed fMRI data released by these datasets[11,18,20–23], with the ROIs they defined. A brief overview of the three datasets focused on natural images is provided below (see Supplementary Fig. 6 for the DeepSoundRecon dataset).

**Deeprecon dataset.** Five healthy subjects (four males and one female; age range = 25–36 years) with normal or corrected-to-normal vision participated in the experiment. The natural image stimuli were selected from 200 representative categories in the ImageNet dataset (2011, fall release)[39]. The natural training images were 1,200 images taken from 150 object categories, and the natural test images were 50 images taken from the remaining 50 object categories[16,40]. The artificial test image stimuli consisted of 40 combinations of five shapes (square, small frame, large frame, plus and cross signs) and eight colors (red, green, blue, cyan, magenta, yellow, white and black). Subjects fixated on a central fixation point while viewing each stimulus for 8 s. The fMRI data were collected using a 3 T scanner, including 6,000 training samples (1,200 natural images with five repetitions), 1,200 test samples (50 natural images with 24 repetitions) and 800 test samples for generalization evaluation (40 artificial images with 20 repetitions) for each subject. The brain regions V1, V2, V3 and V4 were demarcated using a retinotopic mapping experiment[41,42] in each subject's native brain space. The HVC was defined by conventional functional localizers[43–45]. The whole visual cortex was defined as the combination of the regions V1, V2, V3, V4 and HVC.

**THINGS dataset.** Three healthy subjects (one male and two females; mean age = 25.33 years) with normal or corrected-to-normal vision participated in the experiment. The image stimuli were taken from the THINGS object concept and image database[46]. The training images were 8,640 images taken from 720 representative object concepts, with the first 12 examples per concept, and the test images were 100 separate images taken from the remaining THINGS images. Subjects fixated on a central fixation point while each image was presented for 0.5 s, followed by 4 s of fixation without image stimuli. The fMRI data were collected using a 3 T scanner, including 8,640 training samples (8,640 natural images with one repetition) and 1,200 test samples (100 natural images with 12 repetitions) for each subject. The ROIs were defined based on retinotopic mapping[41,42] and functional localizer experiments[43–45], with visual cortex as the combination of all visual areas.

**Natural Scene Dataset.** Eight healthy subjects (two males and six females; age range = 19–32 years) with normal or corrected-to-normal vision participated in the experiment. The image stimuli were sourced

from the 80 COCO categories within Microsoft's COCO image database[47]. The training images comprised 9,000 images that were mutually exclusive across subjects, and the test images consisted of 100 special images taken from 1,000 images that were shared across subjects. Subjects fixated centrally and performed a long-term continuous recognition task, viewing each image for 3 s with a 1 s gap between images. The fMRI data were collected using a 7 T scanner. Subjects 1, 2, 5 and 7, who completed all experiments, were included in our analysis. Each subject had 24,980 training samples (around 9,000 natural images with three repetitions) and 300 test samples (100 natural images with three repetitions). The visual cortex in our analysis was defined using the Human Connectome Project multi-modal cortical parcellation[48].

## Methods for functional alignment

**Content-loss-based neural code converter.** The content-loss-based neural code converter for each pair of subjects uses a nonlinear multilayer perceptron (MLP) to predict the brain activity patterns of one subject (target) from the brain activity patterns of another subject (source). It has two hidden layers with instance normalization and ReLU functions, and the number of units in each hidden layer is half that of the input layer. The converter $\Phi$ takes a source subject's brain activity pattern $\mathbf{x}_i \in \mathbb{R}^m$, which consists of $m$ voxels' values, and predicts the target subject's brain activity pattern as $\Phi(\mathbf{x}_i) \in \mathbb{R}^n$, which consists of $n$ voxels' values. A pre-trained target decoder then takes the converted brain activity pattern $\Phi(\mathbf{x}_i)$ and predicts the decoded feature pattern of the image stimulus as $\mathbf{d}_{il} = \mathbf{W}_l \Phi(\mathbf{x}_i) + \mathbf{b}_l$, where $\mathbf{d}_{il} \in \mathbb{R}^{d_l}$ is the decoded feature pattern consisting of $d_l$ units' values in the $l$-th DNN layer for the $i$-th image stimulus. $\mathbf{W}_l \in \mathbb{R}^{d_l \times n}$ and $\mathbf{b}_l \in \mathbb{R}^{d_l}$ are the decoding matrix and bias vector, respectively, of the pre-trained decoder. The converter is optimized to make decoded features from the converted brain activity similar to the true features of the source subject's stimulus. It is trained to minimize the objective function

$$\mathcal{L}(\Phi) = \sum_i^N \sum_l^L \eta_l \parallel \mathbf{v}_{il} - (\mathbf{W}_l \Phi(\mathbf{x}_i) + \mathbf{b}_l) \parallel^2, \tag{1}$$

where $\mathbf{v}_{il} \in \mathbb{R}^{d_l}$ represents true units' values in the $l$-th DNN layer for the $i$-th image stimulus, $N$ is the number of training samples and $L$ is the number of DNN layers; $\eta_l$ is the parameter that weighs the contribution of the $l$-th layer, which is set to be $1/\parallel \mathbf{v}_{il} \parallel^2$. Although the nonlinear MLP was the default architecture for the content-loss-based converter, we also used several alternative architectures: a linear MLP with two hidden layers; a residual MLP that integrates a nonlinear MLP with three residual blocks[49]; and a vision transformer (ViT) that modifies the input and output layers of the original architecture used by Beyer and colleagues[50].

The converter training was resolved through an iterative process. Each iteration involves a stochastic decoding strategy applied to all VGG19 layers (refer to the 'DNN models' section in the Methods). Specifically, during each iteration, within each convolutional layer, a feature map was chosen at random, and all its units were subjected to the decoding process. Due to the much fewer units in the fully connected layers, all units from these layers were decoded in each iteration; 1,024 iterations were used to ensure the comprehensive involvement of all DNN units from every layer of the image stimulus during the training stage.

**Brain-loss-based neural code converter.** The brain-loss-based neural code converter consists of a set of regularized linear regression models[11]. It takes a source subject's brain activity pattern $\mathbf{x}_i \in \mathbb{R}^m$ consisting of $m$ voxels' values, and predicts the target brain activity pattern as $\hat{\mathbf{y}}_i = \mathbf{M}\mathbf{x}_i + \mathbf{c}$, consisting of $n$ voxels' values. $\mathbf{M} \in \mathbb{R}^{n \times m}$ is the conversion matrix and $\mathbf{c} \in \mathbb{R}^n$ is the bias vector. The converter is trained to minimize the objective function

$$\mathcal{L}(\mathbf{M}, \mathbf{c}) = \sum_i^N \parallel \mathbf{y}_i - (\mathbf{M}\mathbf{x}_i + \mathbf{c}) \parallel^2 + \lambda \parallel \mathbf{M} \parallel_{\mathrm{F}}^2, \tag{2}$$

where $\mathbf{y}_i$ is the measured target subject's brain activity pattern for the $i$-th sample, $N$ is the number of training samples, $\lambda$ is the regularization parameter that is set as described in Ho and colleagues[11], and $\parallel \cdot \parallel_{\mathrm{F}}$ represents the Frobenius norm.

In addition to the brain-loss-based converter, we also used other standard functional alignment methods for comparison, including the pairwise Procrustes transformation and template-mediated Procrustes transformation (hyperalignment[6]; see Supplementary Fig. 3 for details).

## DNN models

We used the VGG19 DNN model[13] implemented using the Caffe library[51] in the converter training and DNN feature decoding analysis. This model is pre-trained for the 1,000-class object recognition task using the images from ImageNet[39] (the pre-trained model is available at https://github.com/BVLC/caffe/wiki/Model-Zoo). The model consists of 16 convolutional layers and three fully connected layers. All the input images to the model were rescaled to 224 × 224 pixels. Outputs from individual units before rectification were used. The number of units in each layer is as follows: conv1_1 and conv1_2, 3,211,264; conv2_1 and conv2_2, 1,605,632; conv3_1, conv3_2, conv3_3 and conv3_4, 802,816; conv4_1, conv4_2, conv4_3 and conv4_4, 401,408; conv5_1, conv5_2, conv5_3 and conv5_4, 100,352; FC6 and FC7, 4,096; and FC8, 1,000.

For the evaluation of reconstructed images, we used another DNN model, the AlexNet[12] implemented using the Caffe library to extract DNN features from the reconstructed images and the presented image. This model is also pre-trained with images in ImageNet to classify 1,000 object categories (available via https://github.com/BVLC/caffe/tree/master/models/bvlc_alexnet). The model consists of five convolutional layers and three fully connected layers. All of the input images to the model were rescaled to 224 × 224 pixels. The number of units in each layer is as follows: conv1, 290,400; conv2, 186,624; conv3 and conv4, 64,896; conv5, 43,264; FC6 and FC7, 4,096; and FC8, 1,000.

We used the CLIP-ViT model[14] implemented using the PyTorch library[52] to examine different decoding schemes. This model is pre-trained on diverse image–text pairs to link images with text descriptions for a variety of visual tasks without task-specific training (available via https://github.com/openai/CLIP). All the input images to the model were rescaled to 224 × 224 pixels. The number of units used in each layer for the image encoder is as follows: conv1, 37,632; transformer blocks 0–11, 38,400 each; ln_post, 768; and model_output, 512.

## Evaluation of conversion accuracy

We evaluated the conversion accuracy using two metrics: pattern correlation and profile correlation. Pattern correlation is the Pearson correlation coefficient between the converted and measured voxel patterns for a test stimulus. Profile correlation is the Pearson correlation coefficient between the sequences of converted and measured responses of a single voxel to the 50 natural test stimuli. Repeated measures of the brain responses to an identical stimulus in fMRI data are subject to measurement noise, which impacts the evaluation of conversion accuracy. To address this issue, we performed the noise ceiling estimation[53,54]. The noise ceiling is calculated by averaging the correlation coefficients between repeated responses to identical stimuli, which reflects the maximum performance that the converter model can achieve given measurement noise in fMRI data. We excluded samples or voxels if their noise ceilings fell below a specified threshold (the 99th percentile of the distribution from random pairs). The obtained correlation coefficients were then normalized by dividing the raw values by their respective noise ceilings. For each individual pair and each ROI, the normalized pattern correlation coefficients for all test stimuli

were averaged to calculate the mean conversion accuracy (pattern), whereas the normalized profile correlation coefficients for all voxels were averaged to calculate the mean conversion accuracy (profile). In the DNN feature decoding and image reconstruction analyses, we retained all voxels to avoid potential information leakage.

## DNN feature decoding analysis

We used a ridge regression model as a DNN feature decoder. This model predicts the feature values of the stimulus, given an fMRI activity pattern evoked by the stimulus. We normalized both the feature values and voxel responses before model training and used a voxel selection procedure. This procedure involved calculating the Pearson correlation coefficients between sequences of voxel responses and feature values for all voxels. The 500 voxels exhibiting the highest absolute correlations were selected for training. We set the ridge regularization parameter to 100 to enhance model robustness. The feature decoding analysis is detailed in the studies by Horikawa and Kamitani[16,20], and Shen and colleagues[18]. For testing the trained decoders, we used the average fMRI pattern over repetitions to improve the signal-to-noise ratio of the fMRI signal. We evaluated the decoding accuracy using pattern correlation and profile correlation. Pattern correlation is the Pearson correlation coefficient between the decoded feature patterns and the true feature patterns for a test stimulus, whereas profile correlation calculates the Pearson correlation coefficient between the sequences of the decoded and true feature values across all test stimuli for a single DNN unit. For each individual pair or each subject (Within), the pattern correlation coefficients for all test stimuli were averaged to calculate the mean decoding accuracy (pattern), whereas the profile correlation coefficients for all DNN units were averaged to calculate the mean decoding accuracy (profile). These mean accuracies were used as data points in group analysis.

## Visual image reconstruction

The reconstruction method used in this study was extended from our original deep image reconstruction study[18]. The pixel values of an input image were optimized to make its image features match the decoded features from brain activity. Following a work by Shen and colleagues[18], we used the feature values before the rectification operation from eight layers (conv1–5 and all fully connected layers). We applied the same natural image prior and extended the loss function by adding a Deep Image Structure and Texture Similarity (DISTS) loss component[55], which uses spatial characteristics of feature maps to improve the detail of image reconstructions. Given the decoded features from multiple layers, an image was reconstructed by solving the following optimization problem:

$$\mathbf{z}^* = \underset{\mathbf{z}}{\arg\min} \left( \mathcal{L}_{\text{mse}}(\mathbf{z}) + \lambda_{\text{tex}} \, \mathcal{L}_{\text{tex}}(\mathbf{z}) + \lambda_{\text{str}} \, \mathcal{L}_{\text{str}}(\mathbf{z}) \right), \quad (3)$$

where $\mathbf{z}$ is the latent vector, and $\mathcal{L}_{\text{mse}}(\mathbf{z})$ is the loss originally used in Shen and colleagues[18]:

$$\mathcal{L}_{\text{mse}}(\mathbf{z}) = \sum_{l}^{L} \gamma_l \| \Psi_l(G(\mathbf{z})) - \mathbf{u}_l \|^2, \quad (4)$$

where $G$ is the deep generator network to enhance the naturalness of the image[56], and the reconstructed image is obtained as $G(\mathbf{z}^*)$; $\Psi_l$ is the function that maps the image to the DNN feature vector of the $l$-th layer; $\mathbf{u}_l \in \mathbb{R}^{P_l \times Q_l \times K_l}$ represents the decoded DNN feature vector of the image at the $l$-th layer, where $P_l$, $Q_l$ and $K_l$ denote the width, height and number of channels, respectively, of the feature maps in the $l$-th layer; $\gamma_l$ is the parameter that weights the contribution of the $l$-th layer and is set to be $1/\|\mathbf{u}_l\|^2$; $L$ is the number of DNN layers. The $\mathcal{L}_{\text{tex}}(\mathbf{z})$ term represents the texture similarity loss, whereas $\mathcal{L}_{\text{str}}(\mathbf{z})$ represents the structure loss; together, they constitute the DISTS loss. The $\lambda_{\text{tex}}$ and $\lambda_{\text{str}}$ terms serve as the coefficients of weights for the texture and structural similarity losses, respectively.

If we denote $\hat{\mathbf{u}}_l = \Psi_l(G(\mathbf{z})) \in \mathbb{R}^{P_l \times Q_l \times K_l}$, then the texture similarity $\mathcal{L}_{\text{tex}}(\mathbf{z})$ and $\mathcal{L}_{\text{str}}(\mathbf{z})$ are defined as follows:

$$\mathcal{L}_{\text{tex}}(\mathbf{z}) = -\sum_{l}^{L} \alpha_l \frac{1}{K_l} \sum_{k} \frac{\mu_k(\mathbf{u}_l) \mu_k(\hat{\mathbf{u}}_l) + \epsilon}{\mu_k(\mathbf{u}_l)^2 + \mu_k(\hat{\mathbf{u}}_l)^2 + \epsilon}, \quad (5)$$

$$\mathcal{L}_{\text{str}}(\mathbf{z}) = -\sum_{l}^{L} \beta_l \frac{1}{K_l} \sum_{k} \frac{\delta_k(\mathbf{u}_l, \hat{\mathbf{u}}_l) + \epsilon}{\delta_k(\mathbf{u}_l) + \delta_k(\hat{\mathbf{u}}_l) + \epsilon}, \quad (6)$$

where

$$\mu_k(\mathbf{u}_l) = \frac{1}{P_l Q_l} \sum_{p_l, q_l} \mathbf{u}_{p_l, q_l, k_l}, \quad (7)$$

$$\delta_k(\mathbf{u}_l) = \frac{1}{P_l Q_l} \sum_{p_l, q_l} (\mathbf{u}_{p_l, q_l, k_l} - \mu_k(\mathbf{u}_l))^2, \quad (8)$$

$$\delta_k(\mathbf{u}_l, \hat{\mathbf{u}}_l) = \frac{1}{P_l Q_l} \sum_{p_l, q_l} \mathbf{u}_{p_l, q_l, k_l} \hat{\mathbf{u}}_{p_l, q_l, k_l} - \mu_k(\mathbf{u}_l) \mu_k(\hat{\mathbf{u}}_l). \quad (9)$$

Here, $\mathbf{u}_{p_l, q_l, k_l} \in \mathbb{R}$ represents the activity value of the $(p,q,k)$-th element for the $l$-layer. The $\alpha_l$ and $\beta_l$ terms are hyperparameters that signify the weight assigned to each layer; these were tuned using data from a subject excluded from the result analyses. A small positive constant $\epsilon$ is included to avoid numerical instability when the denominator is close to zero. We solved the optimization problem using stochastic gradient descent with momentum with 200 iterations.

## Identification analysis

We used pairwise identification analysis to quantify the accuracy of image reconstruction. For each reconstructed image, its features were reshaped into a one-dimensional vector and compared with the true feature vector of the presented image and the false alternative of another image. An identification was considered correct when the correlation coefficient of the reconstructed image's feature vector was greater with the true feature vector than with the false alternative. The false alternative image was sourced from the remaining images in the test dataset. For each reconstructed image, this process was repeated for all remaining test images as the false alternative. For Deeprecon natural images, 49 false alternatives were used per reconstructed image (50 images × 49 comparisons = 2,450 comparisons). For THINGS or NSD images, 99 false alternatives were used per reconstructed image (100 images × 99 comparisons = 9,900 comparisons for each dataset). We defined the identification accuracy for a reconstructed image as the ratio of correct identifications. For each individual pair or each subject (Within), we calculated the mean identification accuracy by averaging the identification accuracies of all reconstructed images, which was used as a data point for group analysis.

## Group statistics

We performed group-level statistical inference by showing the group means and confidence intervals in the figures. For DNN feature decoding and image reconstruction in within-individual analyses, each data point represents the mean accuracy for a subject. The mean accuracies from all subjects were used to calculate the group mean and its 95% confidence interval. For conversion accuracy, DNN feature decoding, and image reconstruction in inter-individual analyses, each data point represents the mean accuracy corresponding to a unique pair of subjects. Due to potential correlations between pairs involving the same subject (dyadic dependency), we applied bootstrapping to these dyadic data points to calculate the group mean and 95% confidence interval. We performed bootstrap sampling separately on the source and target IDs, obtaining data points of the original sample size while ensuring

that pairs where the source and target IDs were the same were excluded. The obtained data points were then used to calculate the mean. This process was repeated 1,000 times to generate bootstrap replicates, from which we calculated the 95% confidence interval.

### Reporting summary

Further information on research design is available in the Nature Portfolio Reporting Summary linked to this article.

## Data availability

The experimental data that support the findings of this study are available publicly. The Deeprecon dataset[57], used for subjects 1–3, is available at https://doi.org/10.18112/openneuro.ds001506.v1.3.1; the training natural image session dataset, used for subjects 4 and 5, is available at https://doi.org/10.18112/openneuro.ds003430.v1.2.0; and the test natural- and artificial-image sessions dataset, used for subjects 4 and 5, is available at https://doi.org/10.18112/openneuro.ds003993.v1.0.0. The THINGS dataset is available at https://doi.org/10.18112/openneuro.ds004192.v1.0.5 (ref. 58); the Natural Scene Dataset is available at https://naturalscenesdataset.org (ref. 22); and the DeepSoundRecon dataset is available at https://doi.org/10.6084/m9.figshare.23633751.v9 (ref. 59). Source data are provided with this paper.

## Code availability

The code that supports the findings of this study is available via Zenodo at https://zenodo.org/records/14910040 (ref. 60), as well as via our repository at https://github.com/KamitaniLab/InterSiteNeuralCodeConversion. Data analysis was performed using Python v.3.8.18.

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

## Acknowledgements

We thank our laboratory team, especially Y. Nagano, K. Shirakawa, E. Doi and H. Izumi, for their invaluable comments and suggestions on the manuscript. This study was supported by Japan Society for the Promotion of Science (KAKENHI grant nos. JP20H05705 and JP20H05954 to Y.K.), the New Energy and Industrial Technology Development Organization (grant no. JPNP20006 to Y.K.), Guardian Robot Project, RIKEN, Japan Agency for Medical Research and Development (grant no. JP24wm0625409 to Y.K.), and Japan Science and Technology Agency (CREST grant nos. JPMJCR18A5 and JPMJCR22P3 to Y.K.).

## Author contributions

H.W. and Y.K. conceptualized the work. H.W. and Y.K. designed the methodology. H.W. and S.C.A. wrote the software. H.W. performed formal analyses. H.W., J.K.H., F.L.C. and J.-Y.P. performed the investigations. H.W. and Y.K. wrote the original draft of the manuscript, whereas H.W., J.K.H., F.L.C., S.C.A., M.T. and Y.K. reviewed and edited it. H.W. and Y.K. visualized the data. Y.K. supervised the project and acquired funding.

## Competing interests

Advanced Telecommunications Research Institute International (ATR) and Honda Motor Company hold a patent (no. US9020586B2), which covers the foundational concept of neural code conversion; Y.K. is one of the inventors of this patent.

## Additional information

**Correspondence and requests for materials** should be addressed to Haibao Wang or Yukiyasu Kamitani.

**Peer review information** *Nature Computational Science* thanks Katharina Dobs, Sreejan Kumar and Feilong Ma for their contribution to

the peer review of this work. Primary Handling Editor: Ananya Rastogi, in collaboration with the *Nature Computational Science* team. Peer reviewer reports are available.

# Reporting Summary

## Statistics

For all statistical analyses, confirm that the following items are present in the figure legend, table legend, main text, or Methods section.

| n/a | Confirmed | |
|---|---|---|
| ☐ | ☒ | The exact sample size (*n*) for each experimental group/condition, given as a discrete number and unit of measurement |
| ☒ | ☐ | A statement on whether measurements were taken from distinct samples or whether the same sample was measured repeatedly |
| ☐ | ☒ | The statistical test(s) used AND whether they are one- or two-sided<br>*Only common tests should be described solely by name; describe more complex techniques in the Methods section.* |
| ☒ | ☐ | A description of all covariates tested |
| ☒ | ☐ | A description of any assumptions or corrections, such as tests of normality and adjustment for multiple comparisons |
| ☐ | ☒ | A full description of the statistical parameters including central tendency (e.g. means) or other basic estimates (e.g. regression coefficient) AND variation (e.g. standard deviation) or associated estimates of uncertainty (e.g. confidence intervals) |
| ☐ | ☒ | For null hypothesis testing, the test statistic (e.g. *F*, *t*, *r*) with confidence intervals, effect sizes, degrees of freedom and *P* value noted<br>*Give P values as exact values whenever suitable.* |
| ☒ | ☐ | For Bayesian analysis, information on the choice of priors and Markov chain Monte Carlo settings |
| ☒ | ☐ | For hierarchical and complex designs, identification of the appropriate level for tests and full reporting of outcomes |
| ☐ | ☒ | Estimates of effect sizes (e.g. Cohen's *d*, Pearson's *r*), indicating how they were calculated |

*Our web collection on statistics for biologists contains articles on many of the points above.*

## Software and code

Policy information about availability of computer code

| Data collection | As this study did not involve new data collection, no software was used to collect data. |
|---|---|
| Data analysis | The data analysis code is written as custom Python scripts (version 3.8.18) and uses the PyTorch library (version 1.7.1) and the Caffe library (version 1.0). The scripts are released publicly on Github (https://github.com/KamitaniLab/InterSiteNeuralCodeConversion). |

For manuscripts utilizing custom algorithms or software that are central to the research but not yet described in published literature, software must be made available to editors and reviewers. We strongly encourage code deposition in a community repository (e.g. GitHub). See the Nature Portfolio guidelines for submitting code & software for further information.

## Data

Policy information about availability of data

All manuscripts must include a data availability statement. This statement should provide the following information, where applicable:

- Accession codes, unique identifiers, or web links for publicly available datasets
- A description of any restrictions on data availability
- For clinical datasets or third party data, please ensure that the statement adheres to our policy

The data used is publicly available and the sources are as follows:
Deeprecon dataset: https://doi.org/10.18112/openneuro.ds001506.v1.3.1 for subjects 1-3, https://doi.org/10.18112/openneuro.ds003430.v1.2.0 for the dataset of training natural-image session for subjects 4 and 5, and https://doi.org/10.18112/openneuro.ds003993.v1.0.0 for the dataset of test natural-image and artificial-

image sessions for subjects 4 and 5;
THINGS dataset: https://doi.org/10.18112/openneuro.ds004192.v1.0.5;
Natural Scene Dataset:https://naturalscenesdataset.org;
DeepSoundRecon dataset: https://doi.org/10.6084/m9.figshare.23633751.v9.

# Human research participants

Policy information about studies involving human research participants and Sex and Gender in Research.

| | |
|---|---|
| Reporting on sex and gender | The Deeorecon dataset recruited five subjects (four males and one female).<br>The THINGS dataset recruited three subjects (one male and two females).<br>The NSD dataset recruited eight subjects (two males and six females).<br>The DeepSoundRecon dataset recruited five subjects (four males and one female). |
| Population characteristics | The Deeprecon dataset: age range 25-36 years, subjects with normal or corrected-to-normal vision.<br>The THINGS dataset: mean age: 25.33 years, subjects with normal or corrected-to-normal vision.<br>The NSD dataset: age range, 19–32 years subjects with normal or corrected-to-normal vision.<br>The DeepSoundRecon dataset: mean age: 27.6 years, subjects with normal hearing. |
| Recruitment | Participants were from publicly available datasets, including Deeprecon, THINGS, NSD, and DeepSoundRecon, all approved by their respective institutional review boards. Informed consent was obtained from all participants prior to data collection. Each dataset has specific inclusion criteria, such as normal or corrected-to-normal vision and no history of neurological or psychiatric disorders (NSD). A potential selection bias may exist, as participants in neuroimaging studies tend to be highly motivated and experienced with experimental tasks, which may impact generalizability. |
| Ethics oversight | The Deeprecon dataset: approved by the Ethics Committee of the Advanced Telecommunications Research Institute International (ATR).<br>The THINGS dataset: approved by the NIH Institutional Review Board.<br>The NSD dataset: approved by the University of Minnesota Institutional Review Board.<br>The DeepSoundRecon dataset: approved by the Ethics Committee of the Advanced Telecommunications Research Institute International (ATR).<br>Our study is also approved by the Ethics Committee of the Advanced Telecommunications Research Institute International (ATR). |

Note that full information on the approval of the study protocol must also be provided in the manuscript.

# Field-specific reporting

Please select the one below that is the best fit for your research. If you are not sure, read the appropriate sections before making your selection.

☒ Life sciences ☐ Behavioural & social sciences ☐ Ecological, evolutionary & environmental sciences

For a reference copy of the document with all sections, see nature.com/documents/nr-reporting-summary-flat.pdf

# Life sciences study design

All studies must disclose on these points even when the disclosure is negative.

| | |
|---|---|
| Sample size | The sample size was predetermined by the original dataset providers, as this study used publicly available datasets. These datasets provide 114 subject pairs for our analysis. Power analysis indicates that a minimum of 15 subject pairs is required to detect a large effect size (Cohen's h = 0.8) with 80% power at α = 0.05. The current sample size exceeds this requirement, ensuring robust statistical power for our analyses. |
| Data exclusions | No data were excluded from the analysis. |
| Replication | The analyses are performed for each subject pair. The effects are replicated across 114 subject pairs. |
| Randomization | In the analysis of the effect of stimulus overlap between converter and decoder trainings, we randomly divided the training samples from the Deeprecon dataset into two distinct halves based on the categories of stimuli. The source subject were provided with 3000 training samples (600 images from 75 randomly selected categories out of 150 categories, with five repetitions of each image), and the target subject were given a different set of 3000 training samples (the remaining 600 images with five repetitions each). This strategy was designed to prevent overlapping stimuli between the source and target subjects and to avoid any pairing in their brain activity patterns. |
| Blinding | Blinding was not applicable in this study, as the data were obtained from publicly available datasets, and the investigators were not involved in participant recruitment, data collection, or group allocation. |

# Reporting for specific materials, systems and methods

We require information from authors about some types of materials, experimental systems and methods used in many studies. Here, indicate whether each material, system or method listed is relevant to your study. If you are not sure if a list item applies to your research, read the appropriate section before selecting a response.

## Materials & experimental systems

| n/a | Involved in the study |
|---|---|
| ☒ | ☐ Antibodies |
| ☒ | ☐ Eukaryotic cell lines |
| ☒ | ☐ Palaeontology and archaeology |
| ☒ | ☐ Animals and other organisms |
| ☒ | ☐ Clinical data |
| ☒ | ☐ Dual use research of concern |

## Methods

| n/a | Involved in the study |
|---|---|
| ☒ | ☐ ChIP-seq |
| ☒ | ☐ Flow cytometry |
| ☐ | ☒ MRI-based neuroimaging |

## Magnetic resonance imaging

### Experimental design

**Design type**

The Deeorecon dataset: passiave image-viewing task
The THINGS dataset: passiave image-viewing task
The NSD dataset: passiave image-viewing task
The DeepSoundRecon dataset: passiave sound-listening task

**Design specifications**

The Deeorecon dataset:  Each presentation of an image lasted for 8 s in a stimulus block. fMRI signals were measured while subjects each viewed 1,290 visual images (8,000 trials) over the course of 15–20 scan sessions.
The THINGS dataset: Each image was presented for 0.5 ms, followed by 4 s of eye fixation without image stimuli. fMRI signals were measured while subjects each viewed 8,740 unique visual images (11,040 trials) over the course of 15–16 scan sessions.
The NSD dataset: Images were presented for 3 s with 1-s gaps in between images. fMRI signals were measured while subjects each viewed 9,000–10,000 distinct natural scenes (22,000–30,000 trials) over the course of 30–40 scan sessions.
The DeepSoundRecon dataset: Each sound stimulus was presented for 8 s within a stimulus block. fMRI signals were measured while subjects each listend 1,250 sound stimuli (5,200 trials) over the course of 13–17 scan sessions.

**Behavioral performance measures**

Not applicable because the passive image-viewing and sound-listening tasks do not involve measurable behavioral performance data.

### Acquisition

**Imaging type(s)**

All imaging types used in this study were functional MRI.

**Field strength**

The Deeorecon dataset: 3T
The THINGS dataset: 3T
The NSD dataset: 7T
The DeepSoundRecon dataset: 3T

**Sequence & imaging parameters**

The Deeorecon dataset: An interleaved T2-weighted gradient-echo echo planar imaging (EPI) scan was performed to acquire functional images covering the entire brain (TR, 2000 ms; TE, 43 ms; flip angle, 80 deg; FOV, 192 × 192 mm; voxel size, 2 × 2 × 2 mm; slice gap, 0 mm; number of slices, 76).
The THINGS dataset: The whole-brain functional MRI data was collected  with 2 mm isotropic resolution (60 axial slices; 2 mm slice thickness; no slice gap; matrix size 96×96; FOV, 192 × 192 mm; TR, 1.5 s; TE, 33ms; flip angle, 75 deg).
The NSD dataset: The primary fMRI sequence involved gradient-echo EPI, FOV 216 mm x 216 mm, matrix size 120 x 120, slice thickness 1.8 mm, orientation axial, TR 1.6 s, TE 22.0 ms, and flip angle 62 deg.
The DeepSoundRecon dataset: An interleaved T2-weighted gradient-echo echo planar imaging (EPI) scan was performed to acquire functional images covering the entire brain (TR, 2000 ms; TE, 43 ms; flip angle, 80 deg; FOV, 192 × 192 mm; voxel size, 2 × 2 × 2 mm; slice gap, 0 mm; number of slices, 76).

**Area of acquisition**

Whole-brain scans

**Diffusion MRI**     ☐ Used     ☒ Not used

### Preprocessing

**Preprocessing software**

Already preprocessed MRI data were used, based on the preprocessing pipeline of Deeprecon dataset, THINGS dataset, NSD dataset, and DeepSoundRecon dataset. Preprocessing involved tools of fMRIPrep, FreeSurfer6, and selected tools from SPM,

FSL, ANTs, and MRTrix3.

| Normalization | Already normalized MRI data were used, based on the preprocessing pipelines of the DeepRecon dataset, THINGS dataset, NSD dataset, and DeepSoundRecon dataset. These data involved subject-native space and atlas spaces (MNI, fsaverage). |

| Normalization template | For data in atlas spaces, Normalization template involved the MNI152 and fsaverage templates. |

| Noise and artifact removal | Already preprocessed MRI data were used, based on the preprocessing pipelines of the DeepRecon dataset, THINGS dataset, NSD dataset, and DeepSoundRecon dataset. For Deeprecon data, the BOLD time series were temporally shifted by 4 s to account for hemodynamic delays and then regressed for nuisance variables. The data samples were finally despiked to reduce extreme values (beyond ±3 SD for each run) in the time series and averaged within each 8-s trial (four volumes). For THINGS data, the ICA denoising was performed, followed by the GLMdenoise method. For the GLM preparation of the NSD data, the data-driven analysis method GLMdenoise and the statistical technique of ridge regression were used. For DeepSoundRecon data, the BOLD time series were temporally shifted by 2 s to account for hemodynamic delays and then regressed for nuisance variables. |

| Volume censoring | No volume censoring was performed. |

## Statistical modeling & inference

| Model type and settings | Predictive model (i.e. decoding model that predicts representation of the stimulus as a function of brain recordings) |

| Effect(s) tested | We test whether fMRI activities is predictive of representations of the stimulus |

Specify type of analysis:   ☐ Whole brain   ☒ ROI-based   ☐ Both

| Anatomical location(s) | The regions of interest (ROIs) were defined based on previous literature (Engel et al., 1994; Sereno et al., 1995; Kourtzi and Kanwisher, 2000; Kanwisher et al., 1997; Epstein and Kanwisher, 1998; Glasser et al., 2016), and the ROI masks were obtained from the dataset. |

| Statistic type for inference<br>(See Eklund et al. 2016) | Prediction performance was evaluated using Pearson correlation between predicted and actual image features.<br>The mean Pearson correlation and its 95% confidence interval (CI) were computed using bootstrap resampling (1,000 iterations). |

| Correction | No multiple comparison correction was applied, as statistical inference was based on the 95% confidence interval (CI) obtained via bootstrap resampling. |

## Models & analysis

| n/a | Involved in the study |
| --- | --- |
| ☒ | ☐ Functional and/or effective connectivity |
| ☒ | ☐ Graph analysis |
| ☐ | ☒ Multivariate modeling or predictive analysis |

| Multivariate modeling and predictive analysis | Feature extraction: The image stimuli are input into different pre-trained DNN models (including VGG19, AlexNet, and CLIP) to obtain the latent DNN features for decoder training, converter training, and evaluation.<br>Model: The neural code converter for each pair of subjects uses a nonlinear Multi-Layer Perceptron (MLP) to predict the brain activity patterns of one subject (target) from the brain activity patterns of another subject (source).<br>Training: The target subject's training data is used to pre-train the target decoder, whereas the source subject's training data is used for the converter training. The converter is optimized so that the converted brain activity is decoded into content representations that closely resemble that of the stimulus given to the source subject.<br>Evaluation: We evaluted the neural code converter model using three metrics: conversion accuracy, decoding accuracy, and image reconstruction. For conversion accuracy, we computed the Pearson correlation between the predicted brain activities and the measured brain activities for each converter model. For decoding accuracy, we calculated the Pearson correlation between the DNN features decoded from the converted brain activities and the true features from the image stimuli. For image reconstruction, we reconstruced images from the converted brain activities and used identification analysis to measure reconstruction accuracy. This approach involved the identification of the presented image out of two alternatives based on the Pearson correlation of image features, including pixel values and DNN features. |

