## [Peer Review file · Nature Computational Science]

Inter-individual and inter-site neural code conversion without shared stimuli

Corresponding Author: Mr Haibao Wang

Version 0:

Decision Letter:

** Please ensure you delete the link to your author homepage in this e-mail if you wish to forward it to your co-authors. **

Dear Mr Wang,

Your manuscript "Inter-individual and inter-site neural code conversion without shared stimuli" has now been seen by 3 referees, whose comments are appended below. You will see that while they find your work of interest, they have raised points that need to be addressed before we can make a decision on publication.

The referees' reports seem to be quite clear. Naturally, we will need you to address *all* of the points raised.

While we ask you to address all of the points raised, the following points need to be substantially worked on:

- While the overall conversion accuracy is comparable between the two methods, the variability across subject pairs appears larger in the content loss method compared to the brain loss method. Please check whether this difference in variability is statistically significant.
- To further understand the variability in conversion accuracy, it would be interesting to examine the correlation of pairwise accuracies between the two methods.
- Please clarify the discrepancy between the decoding and image reconstruction results in Figure 3.
- Please apply the method to measure the alignment between different brain areas within or between subjects.
- Please explore if the model works similarly well when only using the HVC.
- Please show some evidence that the current work's methodology can extend to other domains beyond pure visual perception.

Please use the following link to submit your revised manuscript and a point-by-point response to the referees' comments (which should be in a separate document to any cover letter):

Link Redacted

** This url links to your confidential homepage and associated information about manuscripts you may have submitted or be reviewing for us. If you wish to forward this e-mail to co-authors, please delete this link to your homepage first. **

To aid in the review process, we would appreciate it if you could also provide a copy of your manuscript files that indicates your revisions by making use of Track Changes or similar mark-up tools. Please also ensure that all correspondence is marked with your Nature Computational Science reference number in the subject line.

In addition, please make sure to upload a Word Document or LaTeX version of your text, to assist us in the editorial stage.

To improve transparency in authorship, we request that all authors identified as 'corresponding author' on published papers create and link their Open Researcher and Contributor Identifier (ORCID) with their account on the Manuscript Tracking System (MTS), prior to acceptance. ORCID helps the scientific community achieve unambiguous attribution of all scholarly contributions. You can create and link your ORCID from the home page of the MTS by clicking on 'Modify my Springer Nature account'. For more information please visit please visit <a

<http://www.springernature.com/orcid>>www.springernature.com/orcid.

We hope to receive your revised paper within three weeks. If you cannot send it within this time, please let us know.

Best regards,

Ananya Rastogi, PhD
Senior Editor
Nature Computational Science

Reviewers comments:

Reviewer #1 (Remarks to the Author):

The manuscript by Wang et al. presents an innovative method for aligning functional brain data across different subjects without the need for shared stimuli. This method employs a content loss-based optimization strategy focused on the latent feature space representations (obtained from DNNs) of stimulus content. Initially, a decoder is trained to interpret the target subject's brain data in latent feature space. Next, a converter is optimized to transform the source subject's brain data into that of the target subject. This is achieved by minimizing the loss between the stimulus feature representation (presented to the source subject) and the content latent representation decoded from the converted brain data of the target subject. This setup allows the converter to function without paired brain activity, enabling the use of data from different datasets or entirely without shared stimuli. The results show that this method performs comparably to traditional brain loss-based optimization, which directly trains a converter between the brain activities of the source and the target subjects but requires shared stimuli. Moreover, the authors show successful alignment across recording sites. Overall, the paper is well-written and the methods are clearly described. I have a few questions and suggestions, which I believe would strengthen the manuscript:

1. The authors present profile and pattern correlations between the brain activities of the converted and measured responses on a test set for both the brain loss and content loss methods. While the overall conversion accuracy is comparable between the two methods, the variability across subject pairs appears larger in the content loss method compared to the brain loss method. Could the authors check whether this difference in variability is statistically significant? Assessing the significance of this variability would be important for users to fully evaluate the pros and cons of this novel method.
2. To further understand the variability in conversion accuracy, it would be interesting to examine the correlation of pairwise accuracies between the two methods. Specifically, do pairs that achieve higher alignment in one method also exhibit better alignment using the other method (and vice versa)? Given the recent interest in inter-individual differences in brain alignment (e.g., Broda & de Haas, PNAS, 2024), evaluating whether both methods consistently quantify alignment between pairs of subjects would provide valuable insights.
3. Could the authors help clarify the discrepancy between the decoding and image reconstruction results in Figure 3? If I understand correctly, directly correlating the decoded features of the converted brain activity with those of the measured brain activity showed an advantage for the late layers of the deep neural network (i.e., the fully-connected layers achieved the highest correlation in profile and pattern analyses). However, when reconstructing images from the decoded features of the converted brain activity, the early to mid-level layers of AlexNet produced better results. This suggests that while late-level features are most useful as a latent space for decoding, the similarity between the presented and reconstructed images is higher when using early to mid-level features. The authors discuss the advantage of mid-level layers for reconstruction but not with respect to the decoding analysis. It would be helpful if the authors could comment on this and add a sentence or two explaining this divergence for readers like myself.
4. Regarding the identification accuracy method, I am curious why the authors chose to use this approach to compare the similarity between the original image, the reconstructed image, and two alternatives (in Fig. 3-6). This method seems to heavily depend on the selection of alternative images; if the alternatives are very distinct from the original, the reconstructed image will almost always be identified as more similar. I am concerned that this high performance, approaching a ceiling effect, might obscure actual differences in performance between the methods. It would be helpful if the authors could provide more details in the Methods section explaining how the alternative images were selected. Moreover, based on this approach, two methods can principally both achieve high accuracy, yet one may show a large difference between the presented and reconstructed images, while the other shows a small difference. Therefore, I suggest that the authors present additional quantitative measures, such as the average correlation between the reconstructed and true feature vectors, to more precisely quantify reconstruction quality. Alternatively, they could consider transforming the identification task into a probabilistic decision framework, for example by using Luce's choice rule. These suggestions might enhance the understanding of the similarities and differences between both methods.
5. I am curious whether the authors have considered applying their method to measure the alignment between different brain areas within or between subjects. Specifically, could the converter be adapted to transform neural activity patterns from one cortical area to another? The degree of alignment achieved might reflect the hierarchical processing nature along the visual processing stream. I would be interested to hear the authors' thoughts on the feasibility of this approach and whether it could be incorporated into future studies.

Reviewer #2 (Remarks to the Author):

In the manuscript entitled “Inter-individual and inter-site neural code conversion without shared stimuli,” Wang et al. proposed a new functional alignment method and demonstrated its capability to generalize across stimuli, individuals, and sites. Different from previous methods that aligns one individual to another or aligns a group of individuals to a common template, this method uses the DNN representation as an intermediate. The idea is very interesting, the results are clean, and the manuscript is well written. I enjoyed reading the manuscript.

I feel the current manuscript is already of high quality, and my comments are mainly for clarifications and discussions.

First, I want to point out that not all existing functional alignment methods require shared stimuli, for example, connectivity hyperalignment (Guntupalli et al., 2018). In fact, this method has been used to perform hyperalignment between individuals who watched completely different stimuli (Jiahui et al., 2023, which is also an example of pairwise hyperalignment). Another method, the INT model, factors out stimulus information from stimulus-general neural tuning (Feilong et al., 2023). Though I don't think these existing works will reduce the novelty of the current work, I feel that they should be included in the introduction/discussions for the completeness of related literature.

A key novelty of the method is that it focuses on reconstructing DNN representations from brain activities. My understanding is that what is captured by the model is not the information shared by the source subject and the target subject, but the information shared among the two subjects and the DNN. As is shown by the authors, the shared information is sufficient to reconstruct the image with high fidelity. I wonder if the authors have any speculations on what information is specific to human brains, and what specific to DNN representations?

The model is constructed in a way that it converts source subject representations to target subject representations, then to DNN representations, then to images. The loss function centers on the DNN representations, which is more direct, and it outperforms the brain loss-based models. I am curious if the authors have thought about an even more direct approach, that is, based on image reconstruction loss?

The main results are based on fMRI activity from the whole visual cortex. In Fig. 2, the authors demonstrate that the model well predicts responses in both low- and high-level visual cortex. I wonder if the image reconstruction works similarly well when only using the HVC? I'm asking this because theoretically image reconstruction is trivial if we can accurately measure all neural activity in retina/V1.

I think some of the expressions in the manuscript can be more accurate. For example, it's better to replace “[i]nter-individual variability in fine-grained functional brain organization” with “inter-individual variability in fine-grained functional topographies.” My understanding is that the current method (and similar methods) focuses on the shared functional organization across individuals, and it's just how it instantiates in each brain is different (i.e., topographic idiosyncrasies rather than representational idiosyncrasies).

Reviewer #2 (Remarks on code availability):

If needed, I am happy to review the code, too, given another 1–2 weeks.

Reviewer #3 (Remarks to the Author):

In this manuscript, Wang et al. develop a novel approach to functional alignment based on previous image reconstruction methodologies. In particular, a converter model (initially a two layer perceptron, but they also verified on different architectures like residual MLP and ViTs) is trained to predict brain activity of a target subject using brain activity from a source subject. The way they do this is to pretrain a decoder that uses a Deep Neural Network (DNN) as a latent space to represent brain activity. The decoder is pretrained to convert brain activity of seen stimuli for the target participant into the corresponding DNN layers of the stimulus (in their main experiments, they use VGG). Then, the converter is trained on a “content loss” where the converter's predicted brain activity of the target participant is given to the decoder and the resulting latent DNN features are compared to the ground-truth DNN features of the original image. Importantly, this method does not require the participants to look at shared stimuli, a defining limitation of many previous functional alignment methods. The authors show the utility of their method in functional alignment using the Deeprecon, NSD, and THINGS datasets - three different neuroimaging, natural image visual perception datasets. They showed their method can do functional alignment on participant data collected from different sites (the three datasets differ in particular fMRI scanner used and the specific scanning parameters/conditions). Because of the use of task-optimized DNNs as a latent representation for functional alignment, their method can also allow for other tasks - such as image reconstruction.

Overall, I think the authors have developed an impressive method and rigorous experiments/metrics to showcase its use in functionally aligning neuroimaging data for visual perception tasks on natural images. My main concern is their claim of this as a novel general-purpose functional alignment technique. I believe there should be a significant revision of the manuscript and/or potential follow-up experiments to really understand whether the present work warrants this claim. Below are specific points I think the authors should address:

1. The experiments are restricted to a particular paradigm of visual perception of static images. In particular, even though the

method does not need shared stimuli across participants, on a surface level, it would seem this method requires participants looking at some static images to train the decoder model. Other functional alignment techniques, such as Chen et al. 2015 Shared Response Modeling or Haxby et al. 2020 hyperalignment, necessitate participants watch a common stimulus/movie before doing the experiment, but these methods are commonly used in task paradigms beyond visual perception, like decision making, memory, etc. From the perspective of the experimenter, having their participants watch image stimuli before their experiment that aren't required to be shared is not that much less work than having them watch a common movie before the experiment. From this perspective, the utility of the current work's method might be more restricted to the visual perception domain. If the authors have a different perspective on the potential utility of this method on paradigms outside of visual perception, I would love to hear them and/or potentially specify it in the manuscript.

In my eyes, I think the authors have developed a potentially incredibly cool general-purpose "model-based functional alignment" technique based on task-optimized DNNs, but their experiments are only within a particular paradigm of visual perception of static images. In principle, any task that a neural network can be optimized on can be utilized as the latent representation from which to perform functional alignment. For example, for fMRI data where people are reading or listening to text (Nastase et al. 2021), one can utilize internal layers from foundation models of text (e.g. GPT) or spoken speech (e.g. SpeechT5) to functionally align participants listening or reading different text stimuli. The Centaur model (Binz et al. 2024) is a recent foundation model trained on psychological decision making tasks that was shown to account for brain activity while participants engage in a decision making task during fMRI. DQN models trained to play Atari games can be used to align participants playing reinforcement learning games that have different trajectories (Cross et al. 2020). I think if the authors were able to show some evidence that the current work's methodology can extend to other domains beyond pure visual perception (including but of course not restricted to the examples I outline here), the work would be prove to be useful to any researcher doing neuroimaging (given the versatility of DNNs across many domains). I acknowledge this may seem like a significant amount of work, but I think showing this would make the work incredibly interesting, strong, and useful to a wide variety of researchers. Otherwise, I think it may be necessary for the manuscript to soften the claims of this method being a general-purpose functional alignment method and emphasize its use in the visual perception paradigm, which is definitely a valuable contribution but maybe not as impactful.

2. Related to my points above, it would be informative to show a more traditional functional alignment baseline, such as Shared Response Modeling or Hyperalignment. Even if these end up performing better at some of the metrics used by the authors, that's not necessarily a bad thing since these methods have the added assumption of shared stimuli. Nonetheless, adding these baselines would be informative, especially if this is being pitched as a functional alignment method.

In conclusion, I believe this is exciting work, but will need a major revision, particularly in addressing its improvement beyond other functional alignment methods.

References:

Chen, P. H. C., Chen, J., Yeshurun, Y., Hasson, U., Haxby, J., & Ramadge, P. J. (2015). A reduced-dimension fMRI shared response model. *Advances in neural information processing systems*, 28.

Haxby, J. V., Guntupalli, J. S., Nastase, S. A., & Feilong, M. (2020). Hyperalignment: Modeling shared information encoded in idiosyncratic cortical topographies. *elife*, 9, e56601.

Nastase, S. A., Liu, Y. F., Hillman, H., Zadbood, A., Hasenfratz, L., Keshavarzian, N., ... & Hasson, U. (2021). The "Narratives" fMRI dataset for evaluating models of naturalistic language comprehension. *Scientific data*, 8(1), 250.

Binz, M., Akata, E., Bethge, M., Brändle, F., Callaway, F., Coda-Forno, J., ... & Schulz, E. (2024). Centaur: a foundation model of human cognition. *arXiv preprint arXiv:2410.20268*.

Cross, L., Cockburn, J., Yue, Y., & O'Doherty, J. P. (2021). Using deep reinforcement learning to reveal how the brain encodes abstract state-space representations in high-dimensional environments. *Neuron*, 109(4), 724-738.

Reviewer #3 (Remarks on code availability):

I briefly reviewed the code during my review of the paper and was impressed by its readability and extensive documentation. I commend the authors for sharing it.

Version 1:

Decision Letter:

Our ref: NATCOMPUTSCI-24-1681A

28th January 2025

Dear Dr. Wang,

Thank you for submitting your revised manuscript "Inter-individual and inter-site neural code conversion without shared

stimuli" (NATCOMPUTSCI-24-1681A). It has now been seen by the original referees and their comments are below. The reviewers find that the paper has improved in revision, and therefore we'll be happy in principle to publish it in Nature Computational Science, pending minor revisions to satisfy the referees' final requests and to comply with our editorial and formatting guidelines.

TRANSPARENT PEER REVIEW

Nature Computational Science offers a transparent peer review option for original research manuscripts. We encourage increased transparency in peer review by publishing the reviewer comments, author rebuttal letters and editorial decision letters if the authors agree. Such peer review material is made available as a supplementary peer review file. **Please remember to choose, using the manuscript system, whether or not you want to participate in transparent peer review.**

Thank you again for your interest in Nature Computational Science. Please do not hesitate to contact me if you have any questions.

Sincerely,

Ananya Rastogi, PhD
Senior Editor
Nature Computational Science

ORCID

Reviewer #2 (Remarks to the Author):

The authors have addressed my comments adequately, particularly running the model using only HVC. The results on alignment between brain regions look nice, too.

Regarding the hyperalignment baseline, the ridge hyperalignment method used in the INT model might be a better choice than the classic Procrustes method. However, the manuscript is already very rich in content, and I don't think it's necessary to incorporate it in the current manuscript.

Reviewer #2 (Remarks on code availability):

The code is well written and has proper documentation, which is easy to follow. I haven't got the chance to run the code myself.

In the long run, the authors might want to convert the scripts in these two repositories into a package, which will make it easier to use, further increasing the impact of the work.

Reviewer #3 (Remarks to the Author):

Thank you for addressing my comments. The results on DeepSoundrecon have made me more confident that this method is applicable in more domains, and I am pleased to see that this method has quantitative gains over other functional alignment methods. I have no further comments and look forward to seeing this work in published form.

Sincerely,
Sreejan Kumar
Princeton Neuroscience Institute

Version 2:

Decision Letter:

Dear Mr Wang,

We are pleased to inform you that your Article "Inter-individual and inter-site neural code conversion without shared stimuli" has now been accepted for publication in Nature Computational Science.

Once your manuscript is typeset, you will receive an email with a link to choose the appropriate publishing options for your paper and our Author Services team will be in touch regarding any additional information that may be required.

Authors may need to take specific actions to achieve [compliance with funder and institutional open access mandates](https://www.springernature.com/gp/open-research/funding/policy-compliance-faqs). If your research is supported by a funder that requires immediate open access (e.g. according to [Plan S principles](https://www.springernature.com/gp/open-research/plan-s-compliance)) then you should select the gold OA route, and we will direct you to the compliant route where possible. For authors selecting the subscription publication route, the journal's standard licensing terms will need to be accepted, including [self-archiving policies](https://www.springernature.com/gp/open-research/policies/journal-policies). Those licensing terms will supersede any other terms that the author or any third party may assert apply to any version of the manuscript.

Acceptance of your manuscript is conditional on all authors' agreement with our publication policies (see <https://www.nature.com/natcomputsci/for-authors>). In particular your manuscript must not be published elsewhere and there must be no announcement of the work to any media outlet until the publication date (the day on which it is uploaded onto our web site).

Before your manuscript is typeset, we will edit the text to ensure it is intelligible to our wide readership and conforms to house style. We look particularly carefully at the titles of all papers to ensure that they are relatively brief and understandable.

Once your manuscript is typeset, you will receive a link to your electronic proof via email with a request to make any corrections within 48 hours. If, when you receive your proof, you cannot meet this deadline, please inform us at rjsproduction@springernature.com immediately.

If you have queries at any point during the production process then please contact the production team at rjsproduction@springernature.com.

We welcome the submission of potential cover material (including a short caption of around 40 words) related to your manuscript; suggestions should be sent to Nature Computational Science as electronic files (the image should be 300 dpi at 210 x 297 mm in either TIFF or JPEG format). We also welcome suggestions for the Hero Image, which appears at the top of our [home page](http://www.nature.com/natcomputsci); these should be 72 dpi at 1400 x 400 pixels in JPEG format. Please note that such pictures should be selected more for their aesthetic appeal than for their scientific content, and that colour images work better than black and white or grayscale images. Please do not try to design a cover with the Nature Computational Science logo etc., and please do not submit composites of images related to your work. I am sure you will understand that we cannot make any promise as to whether any of your suggestions might be selected for the cover of the journal.

Best regards,

Ananya Rastogi, PhD
Senior Editor
Nature Computational Science

P.S. Click on the following link if you would like to recommend Nature Computational Science to your librarian: https://www.springernature.com/gp/librarians/recommend-to-your-library

** Visit the Springer Nature Editorial and Publishing website at www.springernature.com/editorial-and-publishing-jobs for more information about our career opportunities. If you have any questions please click here.**

Title: Inter-individual and inter-site neural code conversion without shared stimuli

We would like to thank the Editor and reviewers for their valuable comments, which have helped us clarify and substantially improve the manuscript. We have studied these comments carefully and revised our manuscript to address all the issues raised. Please find below detailed point-by-point responses to the comments. The revisions that have been made to the manuscript are highlighted in blue.

Response to Editor

Editor:

While we ask you to address all of the points raised, the following points need to be substantially worked on:

- 1. While the overall conversion accuracy is comparable between the two methods, the variability across subject pairs appears larger in the content loss method compared to the brain loss method. Please check whether this difference in variability is statistically significant.*
- 2. To further understand the variability in conversion accuracy, it would be interesting to examine the correlation of pairwise accuracies between the two methods.*
- 3. Please clarify the discrepancy between the decoding and image reconstruction results in Figure 3.*
- 4. Please apply the method to measure the alignment between different brain areas within or between subjects.*
- 5. Please explore if the model works similarly well when only using the HVC.*
- 6. Please show some evidence that the current work's methodology can extend to other domains beyond pure visual perception.*

Response: We have provided detailed responses to these comments. All of them have been fully addressed with additional results and explanations. We also revised the manuscript according to the reviewers' comments to make these points clear to our readers.

Response to Reviewers

Reviewer: 1

The manuscript by Wang et al. presents an innovative method for aligning functional brain data across different subjects without the need for shared stimuli. This method employs a content loss-based optimization strategy focused on the latent feature space representations (obtained from DNNs) of stimulus content. Initially, a decoder is trained to interpret the target subject's brain data in latent feature space. Next, a converter is optimized to transform the source subject's brain data into that of the target subject. This is achieved by minimizing the loss between the stimulus feature representation (presented to the source subject) and the content latent representation decoded from the converted brain data of the target subject. This setup allows the converter to function without paired brain activity, enabling the use of data from different datasets or entirely without shared stimuli. The results show that this method performs comparably to traditional brain loss-based optimization, which directly trains a converter between the brain activities of the source and the target subjects but requires shared stimuli. Moreover, the authors show successful alignment across recording sites. Overall, the paper is well-written and the methods are clearly described.

Response: We are grateful for your careful reading and positive feedback on our manuscript.

1. The authors present profile and pattern correlations between the brain activities of the converted and measured responses on a test set for both the brain loss and content loss methods. While the overall conversion accuracy is comparable between the two methods, the variability across subject pairs appears larger in the content loss method compared to the brain loss method. Could the authors check whether this difference in variability is statistically significant? Assessing the significance of this variability would be important for users to fully evaluate the pros and cons of this novel method.

Response: Thank you for your suggestions. We added post-hoc Levene's test to check whether this difference in variability is statistically significant in conversion accuracy. We have added the results in the Results section (lines 115–118 in p5), highlighted in blue:

“Although some differences in variance are observed between the two methods, they were not statistically significant (post-hoc Levene's test, pattern correlation: $p > 0.05$ for all ROIs except V2, where $p = 0.039$; profile correlation: $p > 0.05$ for all ROIs).”

2. *To further understand the variability in conversion accuracy, it would be interesting to examine the correlation of pairwise accuracies between the two methods. Specifically, do pairs that achieve higher alignment in one method also exhibit better alignment using the other method (and vice versa)? Given the recent interest in inter-individual differences in brain alignment (e.g., Broda & de Haas, PNAS, 2024), evaluating whether both methods consistently quantify alignment between pairs of subjects would provide valuable insights.*

Response: Thank you for your suggestions. We examined the correlation of pairwise accuracies between the two methods and show the results in Supplementary Fig. 2 for each ROI. We have also added the results in the Results section (lines 118–119 in p5):

“Both methods demonstrated relatively consistent alignment, with individual pairs showing correlated performance between the two methods (Supplementary Fig. 2).”

3. *Could the authors help clarify the discrepancy between the decoding and image reconstruction results in Figure 3? If I understand correctly, directly correlating the decoded features of the converted brain activity with those of the measured brain activity showed an advantage for the late layers of the deep neural network (i.e., the fully-connected layers achieved the highest correlation in profile and pattern analyses). However, when reconstructing images from the decoded features of the converted brain activity, the early to mid-level layers of AlexNet produced better results. This suggests that while late-level features are most useful as a latent space for decoding, the similarity between the presented and reconstructed images is higher when using early to mid-level features. The authors discuss the advantage of mid-level layers for reconstruction but not with respect to the decoding analysis. It would be helpful if the authors could comment on this and add a sentence or two explaining this divergence for readers like myself.*

Response: Thank you for the comment. It is also closely related to your comment 4. We have also added the results based on a correlation analysis of the extracted features from the reconstructions, as shown in Supplementary Fig. 12. As you suggested, this analysis calculates the average correlation between the reconstructed and true feature vectors. The correlation-based results (Supplementary Fig. 12A, B) exhibit a similar tendency across different layers to decoding results (Fig. 3B, C). The discrepancy can arise from the nature of different metrics: successful identification requires features that distinctly differentiate between stimuli, whereas high correlations can occur even when the predictions capture only common patterns shared across stimuli. We speculate that such common components could be pronounced or better predicted at higher layers, resulting in increased correlations. We have also explained this point in the Results section (lines 181–188 in p9):

“We then calculated the mean identification accuracy across all reconstructions for each individual pair, presenting these accuracies at the group level in Fig. 3F, G. Features extracted from reconstructions achieved higher identification accuracy at the mid-level layers, while correlation-based evaluation of decoded features showed higher performance at the high-level layers (Fig. 3B, C). The correlation analysis of extracted features from reconstructions also showed higher predictions at the high-level layers (Supplementary Fig. 12). This discrepancy between identification and correlation metrics can arise because successful identification requires features that distinctly differentiate between stimuli, whereas high correlations can occur even when predictions capture only common patterns across stimuli²³.”

4. Regarding the identification accuracy method, I am curious why the authors chose to use this approach to compare the similarity between the original image, the reconstructed image, and two alternatives (in Fig. 3-6). This method seems to heavily depend on the selection of alternative images; if the alternatives are very distinct from the original, the reconstructed image will almost always be identified as more similar. I am concerned that this high performance, approaching a ceiling effect, might obscure actual differences in performance between the methods. It would be helpful if the authors could provide more details in the Methods section explaining how the alternative images were selected. Moreover, based on this approach, two methods can principally both achieve high accuracy, yet one may show a large difference between the presented and reconstructed images, while the other shows a small difference. Therefore, I suggest that the authors present additional quantitative measures, such as the average correlation between the reconstructed and true feature vectors, to more precisely quantify reconstruction quality. Alternatively, they could consider transforming the identification task into a probabilistic decision framework, for example by using Luce’s choice rule. These suggestions might enhance the understanding of the similarities and differences between both methods.

Response: Thank you for your insightful comments and suggestions. We apologize for the lack of clarity in our initial description. We hope the revised version provides clarity on identification analysis. We have made the following modifications to Methods (lines 725–731 in p24):

“We used pairwise identification analysis to quantify the accuracy of image reconstruction. For each reconstructed image, its features were reshaped into a one-dimensional vector and compared with the true feature vector of the presented image and the false alternative of another image. An identification was considered correct when the correlation coefficient of the reconstructed image’s feature vector was greater with the true feature vector than with the false alternative. The false alternative image was sourced from the remaining images in the test dataset. For each reconstructed image, this process was repeated for all remaining test images as the false alternative. For Deeprecon natural

images, 49 false alternatives were used per reconstructed image (50 images \times 49 comparisons = 2450 comparisons). For THINGS or NSD images, 99 false alternatives were used per reconstructed image (100 images \times 99 comparisons = 9900 comparisons for each dataset). We defined the identification accuracy for a reconstructed image as the ratio of correct identifications. For each individual pair or each subject (Within), we calculated the mean identification accuracy by averaging the identification accuracies of all reconstructed images, which was used as a data point for group analysis.”

We believe that identification accuracy is a practical and relevant metric for many image recognition tasks, as discussed in Comment 3. To provide a more comprehensive comparison between the two methods, we have included additional quantitative measures, as you suggested. Specifically, we conducted a correlation analysis of the extracted features from the reconstructions, and the results are presented in Supplementary Fig. 12. We have also added an explanation of these results in the revised Results section (lines 181–188 on p9). We hope these updates address your concerns and provide additional insights into the similarities and differences between the methods.

“We then calculated the mean identification accuracy across all reconstructions for each individual pair, presenting these accuracies at the group level in Fig. 3F, G. Features extracted from reconstructions achieved higher identification accuracy at the mid-level layers, while correlation-based evaluation of decoded features showed higher performance at the high-level layers (Fig. 3B, C). The correlation analysis of extracted features from reconstructions also showed higher predictions at the high-level layers (Supplementary Fig. 12). This discrepancy between identification and correlation metrics can arise because successful identification requires features that distinctly differentiate between stimuli, whereas high correlations can occur even when predictions capture only common patterns across stimuli²³.”

5. I am curious whether the authors have considered applying their method to measure the alignment between different brain areas within or between subjects. Specifically, could the converter be adapted to transform neural activity patterns from one cortical area to another? The degree of alignment achieved might reflect the hierarchical processing nature along the visual processing stream. I would be interested to hear the authors' thoughts on the feasibility of this approach and whether it could be incorporated into future studies.

Response: Thank you for your suggestion. We agree that exploring the alignment between different brain areas using our method is an intriguing approach, and we have included the preliminary analysis in both the Results and Discussion sections. In the revised Results section, we showed a whole visual cortex (VC) conversion, where the converter was trained to predict activity patterns in the target VC using the entire VC of the source subject, with accuracy evaluated separately for each region of interest (ROI).

We then extended the conversion analysis to different ROI selections. First, we conducted subarea-wise conversion analyses by training separate converters for paired ROIs between the source and target subjects and calculated accuracy for each ROI individually, as shown in Supplementary Fig. 4. This comparison assesses the impact of incorporating prior anatomical information into the conversion process. Next, we conducted subarea-wise conversion analyses between different visual subareas. This analysis involved predicting activity patterns in a target subarea using different source subareas (e.g., using source V1, V2, V3, V4, or HVC individually to predict target V1). The preliminary results are presented in Supplementary Fig. 5. We have included these results in the revised Results section (lines 122–129 on p5):

“Additionally, we extended our analysis to different selections of visual ROIs. We performed subarea-wise conversion analyses by training separate converters for paired ROIs between source and target subjects¹². When using corresponding ROIs, the subarea-wise approach achieved comparable accuracies to the whole VC conversion (see Supplementary Fig. 4 for additional details). When predicting target ROI activity from different source ROIs, the highest conversion accuracies were observed between subareas at similar hierarchical levels (Supplementary Fig. 5). These results indicate that the content loss-based approach can effectively convert brain activity patterns across subjects, preserving hierarchical neural representations.”

We have also discussed the above point in the revised Discussion (lines 495–508 on p18):

“Our approach provides a data-driven framework for exploring functional homology between brain areas through inter-individual neural code conversion. While we demonstrated this using well-characterized visual areas, the method can potentially extend to comparing neural representations between less understood brain regions or even across species. By training converters to predict activity patterns between different source and target regions, we can quantitatively assess their functional correspondence without requiring prior assumptions about their roles. The conversion accuracy between regions serves as an objective measure of their representational similarity. In our analysis of the visual system, this revealed stronger conversion between areas at similar hierarchical levels (Supplementary Fig. 5), consistent with known functional organization. This suggests that our approach may help uncover organizational principles in other neural systems where the functional hierarchy is less clear. The framework may be particularly valuable for establishing cross-species homologies by identifying regions that process information in similar ways, even when their anatomical locations or evolutionary origins differ.”

Reviewer: 2

In the manuscript entitled “Inter-individual and inter-site neural code conversion without shared stimuli,” Wang et al. proposed a new functional alignment method and demonstrated its capability to generalize across stimuli, individuals, and sites. Different from previous methods that aligns one individual to another or aligns a group of individuals to a common template, this method uses the DNN representation as an intermediate. The idea is very interesting, the results are clean, and the manuscript is well written. I enjoyed reading the manuscript. I feel the current manuscript is already of high quality, and my comments are mainly for clarifications and discussions.

Response: We are grateful for your careful reading and positive feedback on our manuscript.

1. First, I want to point out that not all existing functional alignment methods require shared stimuli, for example, connectivity hyperalignment (Guntupalli et al., 2018). In fact, this method has been used to perform hyperalignment between individuals who watched completely different stimuli (Jiahui et al., 2023, which is also an example of pairwise hyperalignment). Another method, the INT model, factors out stimulus information from stimulus-general neural tuning (Feilong et al., 2023). Though I don’t think these existing works will reduce the novelty of the current work, I feel that they should be included in the introduction/discussions for the completeness of related literature.

Response: Thank you for the clarification. We have discussed the above points in the revised Discussion (lines 356–362 in p15):

“[...] While shared stimuli are commonly used in brain loss-based alignment tasks, they may not be strictly necessary in other contexts. For example, connectivity hyperalignment⁵ aligns neural representations using functional connectivity of cortical fields, enabling alignment even when individuals experience different stimuli⁴². Similarly, the Individualized Neural Tuning (INT) model⁴³ separates stimulus-specific information from neural response tuning and captures general neural tuning shared across individuals and generalized across different stimuli. [...]”

2. A key novelty of the method is that it focuses on reconstructing DNN representations from brain activities. My understanding is that what is captured by the model is not the information shared by the source subject and the target subject, but the information shared among the two subjects and the DNN. As is shown by the authors, the shared information is sufficient to reconstruct the image with high fidelity. I wonder if the authors have any speculations on what

information is specific to human brains, and what specific to DNN representations?

Response: Thank you for your insightful question. While our results demonstrate high-fidelity inter-individual image reconstruction, we acknowledge that our choice of VGG19 features as a latent representation of visual content may not fully capture the diverse aspects of visual experience and neural representations. The current DNN representation primarily focuses on 2D visual features, and thus, the converter may fail to align the neural representations of other aspects such as motion perception, 3D perception, and global perception, as well as higher cognitive processes. Further research is needed to understand how the converter trained on a specific content representation generalizes to preserving other types of information and to elucidate the mechanisms of information sharing between brains and DNNs.

We discussed this point in relation to the limitation of using VGG19 features as a latent representation of visual content in the revised Discussion (lines 397–411 in p16):

“However, our choice of VGG19 features as a latent representation of visual content may not fully capture the diverse aspects of visual experience and neural representations. While VGG19 primarily models the ventral visual system’s 2D perception processes, it may inadequately reflect other aspects such as motion perception, 3D perception, and global perception. Different DNN architectures, with their unique designs and training paradigms, might capture distinct aspects of neural representations. [...] While these diverse approaches show promise, determining optimal features for visual content in neural code conversion requires further investigation.”

3. The model is constructed in a way that it converts source subject representations to target subject representations, then to DNN representations, then to images. The loss function centers on the DNN representations, which is more direct, and it outperforms the brain loss-based models. I am curious if the authors have thought about an even more direct approach, that is, based on image reconstruction loss?

Response: Thank you for your thoughtful insights. Using an image reconstruction loss directly is a viable alternative. However, our current reconstruction method relies on many iterative optimization processes, which makes it technically challenging to integrate the reconstruction algorithm directly into the current converter model. If reconstruction methods can simultaneously achieve accuracy, speed, and resource efficiency in image generation, it may become easier to incorporate an image reconstruction loss directly, potentially enhancing the model's performance.

We have discussed this point in the revised Discussion (lines 407–411 in p16):

“[...] It should also be noted that content representation is not limited to model-derived latent features. When combined with reconstruction models, pixel-level representations of images or movies can directly define content loss. While these diverse approaches show promise, determining optimal features for visual content in neural code conversion requires further investigation.”

4. The main results are based on fMRI activity from the whole visual cortex. In Fig. 2, the authors demonstrate that the model well predicts responses in both low- and high-level visual cortex. I wonder if the image reconstruction works similarly well when only using the HVC? I'm asking this because theoretically image reconstruction is trivial if we can accurately measure all neural activity in retina/V1.

Response: Thank you for your question. We also performed image reconstructions from each ROI, under three comparison conditions. Compared to reconstructions from the VC, those from other ROIs show a slight degradation in quality but still reflect the main characteristics of the presented images. We have shown these results in Supplementary Figs. 10, 11, and revised the Results section (lines 170–171 in p7):

“[...] see Supplementary Figs. 10, 11 for the reconstructions from each ROI. [...]”

5. I think some of the expressions in the manuscript can be more accurate. For example, it's better to replace “[i]nter-individual variability in fine-grained functional brain organization” with “inter-individual variability in fine-grained functional topographies.” My understanding is that the current method (and similar methods) focuses on the shared functional organization across individuals, and it's just how it instantiates in each brain is different (i.e., topographic idiosyncrasies rather than representational idiosyncrasies).

Response: Thank you for your suggestion. We have reviewed the manuscript and revised relevant expressions to improve accuracy and clarity. For example, we have updated the expressions as follows:

“Inter-individual variability in fine-grained functional brain organization” has been revised to “Inter-individual variability in fine-grained **functional topographies**.”

Reviewer: 3

In this manuscript, Wang et al. develop a novel approach to functional alignment based on previous image reconstruction methodologies. In particular, a converter model (initially a two layer perceptron, but they also verified on different architectures like residual MLP and ViTs) is trained to predict brain activity of a target subject using brain activity from a source subject. The way they do this is to pretrain a decoder that uses a Deep Neural Network (DNN) as a latent space to represent brain activity. The decoder is pretrained to convert brain activity of seen stimuli for the target participant into the corresponding DNN layers of the stimulus (in their main experiments, they use VGG). Then, the converter is trained on a “content loss” where the converter’s predicted brain activity of the target participant is given to the decoder and the resulting latent DNN features are compared to the ground-truth DNN features of the original image. Importantly, this method does not require the participants to look at shared stimuli, a defining limitation of many previous functional alignment methods. The authors show the utility of their method in functional alignment using the Deeprecon, NSD, and THINGS datasets - three different neuroimaging, natural image visual perception datasets. They showed their method can do functional alignment on participant data collected from different sites (the three datasets differ in particular fMRI scanner used and the specific scanning parameters/conditions). Because of the use of task-optimized DNNs as a latent representation for functional alignment, their method can also allow for other tasks - such as image reconstruction.

Overall, I think the authors have developed an impressive method and rigorous experiments/metrics to showcase its use in functionally aligning neuroimaging data for visual perception tasks on natural images. My main concern is their claim of this as a novel general-purpose functional alignment technique. I believe there should be a significant revision of the manuscript and/or potential follow-up experiments to really understand whether the present work warrants this claim. Below are specific points I think the authors should address:

In conclusion, I believe this is exciting work, but will need a major revision, particularly in addressing its improvement beyond other functional alignment methods.

Response: Thank you for your careful reading and comments. To address the major concern, we have followed your suggestion and incorporated two additional standard functional methods into the manuscript: the pairwise Procrustes transformation and the template-mediated Procrustes transformation (hyperalignment; Haxby, 2011). Additionally, we have included preliminary analyses of auditory neural code conversion using data from the DeepSoundRecon dataset (Park et al., 2023). Below, you will find detailed responses to the comments.

1. The experiments are restricted to a particular paradigm of visual perception of static images. In particular, even though the method does not need shared stimuli across participants, on a surface level, it would seem this method requires participants looking at some static images to train the decoder model. Other functional alignment techniques, such as Chen et al. 2015 Shared Response Modeling or Haxby et al. 2020 hyperalignment, necessitate participants watch a common stimulus/movie before doing the experiment, but these methods are commonly used in task paradigms beyond visual perception, like decision making, memory, etc. From the perspective of the experimenter, having their participants watch image stimuli before their experiment that aren't required to be shared is not that much less work than having them watch a common movie before the experiment. From this perspective, the utility of the current work's method might be more restricted to the visual perception domain. If the authors have a different perspective on the potential utility of this method on paradigms outside of visual perception, I would love to hear them and/or potentially specify it in the manuscript.

In my eyes, I think the authors have developed a potentially incredibly cool general-purpose "model-based functional alignment" technique based on task-optimized DNNs, but their experiments are only within a particular paradigm of visual perception of static images. In principle, any task that a neural network can be optimized on can be utilized as the latent representation from which to perform functional alignment. For example, for fMRI data where people are reading or listening to text (Nastase et al. 2021), one can utilize internal layers from foundation models of text (e.g. GPT) or spoken speech (e.g. SpeechT5) to functionally align participants listening or reading different text stimuli. The Centaur model (Binz et al. 2024) is a recent foundation model trained on psychological decision making tasks that was shown to account for brain activity while participants engage in a decision making task during fMRI. DQN models trained to play Atari games can be used to align participants playing reinforcement learning games that have different trajectories (Cross et al. 2020). I think if the authors were able to show some evidence that the current work's methodology can extend to other domains beyond pure visual perception (including but of course not restricted to the examples I outline here), the work would be prove to be useful to any researcher doing neuroimaging (given the versatility of DNNs across many domains). I acknowledge this may seem like a significant amount of work, but I think showing this would make the work incredibly interesting, strong, and useful to a wide variety of researchers. Otherwise, I think it may be necessary for the manuscript to soften the claims of this method being a general-purpose functional alignment method and emphasize its use in the visual perception paradigm, which is definitely a valuable contribution but maybe not as impactful.

Response: Thank you for your thoughtful comments. We fully acknowledge the importance of other functional alignment approaches and appreciate your suggestion to

extend our method to additional domains. To address this, we have included preliminary analyses of auditory neural code conversion using data from 12 subject pairs in the DeepSoundRecon dataset (Park et al., 2023). These analyses examined the extensibility of our method beyond the visual domain. Detailed information about the data and results is provided in Supplementary Fig. 6. We have also incorporated corresponding descriptions into the Introduction, Results, Discussion, and Methods sections. We believe these revisions further strengthen the applicability of our approach and align with your suggestion.

Revision in the Introduction section (lines 62–63 in p3):

“While our approach is validated in the vision domain, preliminary analysis suggests these principles can be extended to other domains.”

Revision in the Results section:

“Additionally, to examine the extensibility of our method to other domains, we also performed similar neural code conversion analyses within the whole auditory cortex (AC) using the DeepSoundRecon dataset³⁰.” (lines 87–89 in p3)

“While the content loss-based converter was not explicitly trained to align brain activity patterns, its performance measured by correlations between converted and measured brain activity patterns was comparable to, or only slightly lower than, the brain loss-based method across all examined visual subareas.” (lines 112–115 in p5)

“To further evaluate the potential of our approach, we conducted preliminary analyses examining neural responses in the auditory cortex during sound stimulus presentation³⁰. Using the same content loss-based framework, we observed robust inter-individual conversion of auditory neural representations (Supplementary Fig. 6A-E). The performance trends were similar to those observed in the visual domain (Fig. 2E, F), though profile correlations showed a greater decrease in the content loss-based method. The successful conversion of both visual and auditory cortical responses suggests the potential versatility of our approach for studying neural representations across different sensory and cognitive domains.” (lines 130–137 in p5-7)

“Similarly, preliminary auditory decoding analyses within the whole auditory cortex (AC) revealed consistent performance for content loss-based converters (Supplementary Fig. 6F, G).” (lines 161–163 in p7)

According to your comments and suggestions, we also made the revisions in the Discussion section:

“The preliminary result within the auditory domain using our approach further suggests its potential extensibility to other domains.” (lines 324–326 in p14)

“While our primary focus is on the visual perception task, latent representations from other tasks optimized using neural networks may also serve as a basis for functional alignment. Our preliminary analysis showed hierarchical DNN features derived from sound stimuli as latent representations enabled auditory neural code conversion and inter-individual decoding, suggesting the method’s extensibility beyond visual perception (Supplementary Fig.6). The decoded DNN features can be applied to downstream tasks, such as sound reconstruction³⁰. Representations from foundation models, including Generative Pre-trained Transformers (GPTs) for text processing or SpeechT5⁶³ for spoken language, could also be used to align brain activity for semantic tasks, such as reading or listening to text⁶⁴. Similarly, task-specific representations, such as those from the Centaur model⁶⁵ trained on psychological decision-making tasks, and from Deep Q-Network (DQN) models⁶⁶ trained on reinforcement learning environments like Atari games⁶⁷, could be used to align brain activity during decision-making and gameplay paradigms.” (lines 477–488 in p18)

Revision in the Methods section (lines 518–523 in p19):

“We reanalyzed four existing datasets published previously and publicly available^{12, 25, 27–30}. [...] Subjects 13-16 correspond to subjects 2-4 from the DeepSoundRecon dataset³⁰. We used the preprocessed fMRI data released by these datasets^{12,25,27–30}, with the ROIs they defined. A brief overview of the three datasets focused on natural images is provided below (see Supplementary Fig. 6 for the DeepSoundRecon dataset).”

2. Related to my points above, it would be informative to show a more traditional functional alignment baseline, such as Shared Response Modeling or Hyperalignment. Even if these end up performing better at some of the metrics used by the authors, that’s not necessarily a bad thing since these methods have the added assumption of shared stimuli. Nonetheless, adding these baselines would be informative, especially if this is being pitched as a functional alignment method.

Response: Thank you for your comment. We have incorporated additional standard functional alignment methods: the pairwise Procrustes transformation and the template-mediated Procrustes transformation (hyperalignment; Haxby et al., 2011). We performed similar analyses using these methods as we did with the neural code conversion. Details of the methods and analyses are presented in Supplementary Figs. 3, 7. Corresponding descriptions have also been added to the Results and Methods sections.

Revision in the Results section:

“We also tested other standard methods for functional alignment (hyperlignment⁶), showing similar conversion accuracies (see Supplementary Fig. 3 for additional details on functional alignment methods).” (lines 119–121 in p5)

“Other standard functional alignment methods exhibited lower accuracies (Supplementary Fig. 7A, B).” (lines 160–161 in p7)

“The reconstructed images obtained from converters using the brain loss and content loss, as well as from the other functional alignment methods (Supplementary Fig. 7C), captured the essential characteristics and details of the presented images, with visual objects being similarly recognizable to those in the within-individual reconstructions.” (lines 171–174 in p7)

“The other standard functional alignment methods achieved slightly lower accuracies (Supplementary Fig. 7D).” (lines 192–193 in p9)

Revision in the Methods section (lines 623–625 in p21):

“In addition to the brain loss-based converter, we also used other standard functional alignment methods for comparison, including the pairwise Procrustes transformation and template-mediated Procrustes transformation (hyperlignment⁶; See Supplementary Fig.3 for details).”

Remarks on code availability: I briefly reviewed the code during my review of the paper and was impressed by its readability and extensive documentation. I commend the authors for sharing it.

Response: Thank you for taking the time to review the code and for your positive feedback.

Reference :

Park, J. Y., Tsukamoto, M., Tanaka, M., & Kamitani, Y. (2023). Sound reconstruction from human brain activity via a generative model with brain-like auditory features. *arXiv preprint arXiv:2306.11629*.

Haxby, J. V., Guntupalli, J. S., Connolly, A. C., Halchenko, Y. O., Conroy, B. R., Gobbini, M. I., ... & Ramadge, P. J. (2011). A common, high-dimensional model of the representational space in human ventral temporal cortex. *Neuron*, 72(2), 404-416.

Title: Inter-individual and inter-site neural code conversion without shared stimuli

We would like to thank the reviewers for their valuable feedback. We have carefully considered all the comments and have provided detailed point-by-point responses below.

Response to Reviewers

Reviewer: 1

The authors have addressed my comments adequately, particularly running the model using only HVC. The results on alignment between brain regions look nice, too. Regarding the hyperalignment baseline, the ridge hyperalignment method used in the INT model might be a better choice than the classic Procrustes method. However, the manuscript is already very rich in content, and I don't think it's necessary to incorporate it in the current manuscript.

Response: We are grateful for your review and positive feedback on the revised manuscript.

Remarks on code availability: The code is well written and has proper documentation, which is easy to follow. I haven't got the chance to run the code myself. In the long run, the authors might want to convert the scripts in these two repositories into a package, which will make it easier to use, further increasing the impact of the work.

Response: Thank you for your review and positive feedback on the code. We appreciate your suggestion and have now consolidated the scripts into a single repository to enhance usability and accessibility. We also plan to further refine and package our code as part of future developments.

Reviewer: 2

Thank you for addressing my comments. The results on DeepSoundRecon have made me more confident that this method is applicable in more domains, and I am pleased to see that this method has quantitative gains over other functional alignment methods. I have no further comments and look forward to seeing this work in published form.

Response: We are grateful for your positive feedback on the revised manuscript.